# Smart dynamic hybrid membranes with self-cleaning capability

Elvira Pantuso[1,7], Ejaz Ahmed [2,7], Enrica Fontananova[1], Adele Brunetti[1], Ibrahim Tahir [2], Durga Prasad Karothu [2], Nisreen Amer Alnaji[3,4], Ghada Dushaq[4], Mahmoud Rasras[3,4], Panče Naumov [2,3,5,6] ✉ & Gianluca Di Profio [1] ✉

The growing freshwater scarcity has caused increased use of membrane desalination of seawater as a relatively sustainable technology that promises to provide long-term solution for the increasingly water-stressed world. However, the currently used membranes for desalination on an industrial scale are inevitably prone to fouling that results in decreased flux and necessity for periodic chemical cleaning, and incur unacceptably high energy cost while also leaving an environmental footprint with unforeseeable long-term consequences. This extant problem requires an immediate shift to smart separation approaches with self-cleaning capability for enhanced efficiency and prolonged operational lifetime. Here, we describe a conceptually innovative approach to the design of smart membranes where a dynamic functionality is added to the surface layer of otherwise static membranes by incorporating stimuli-responsive organic crystals. We demonstrate a gating effect in the resulting smart dynamic membranes, whereby mechanical instability caused by rapid mechanical response of the crystals to heating slightly above room temperature activates the membrane and effectively removes the foulants, thereby increasing the mass transfer and extending its operational lifetime. The approach proposed here sets a platform for the development of a variety of energy-efficient hybrid membranes for water desalination and other separation processes that are devoid of fouling issues and circumvents the necessity of chemical cleaning operations.

Fresh water is becoming an increasingly scarce commodity due to population growth, industrialization, climate change, and increasing water contamination[1,2]. Currently, about 1.8 billion people have limited access to drinking water[3], and this condition will be inevitably exacerbated over the next decade, concomitant with the growing human population and economies of the developing countries, leading to a predicted 40% shortfall in water supplies by 2030[4]. Desalination could provide a steady and sustainable supply of high-quality fresh water through the treatment of seawater and brackish groundwater[5], which together account for 97.5% of the Earth's global water resources[6]. At

[1]Consiglio Nazionale delle Ricerche (CNR), Istituto per la Tecnologia delle Membrane (ITM), Via P. Bucci, Cubo 17/C, 87036 Rende (CS), Italy. [2]Smart Materials Lab, New York University Abu Dhabi, PO Box 129188 Abu Dhabi, UAE. [3]Center for Smart Engineering Materials, New York University Abu Dhabi, PO Box 129188 Abu Dhabi, UAE. [4]Division of Engineering, New York University Abu Dhabi, PO Box 129188 Abu Dhabi, UAE. [5]Research Center for Environment and Materials, Macedonian Academy of Sciences and Arts, Bul. Krste Misirkov 2, MK–1000, Skopje, Macedonia. [6]Molecular Design Institute, Department of Chemistry, New York University, 100 Washington Square East, New York, NY 10003, USA. [7]These authors contributed equally: Elvira Pantuso, Ejaz Ahmed. ✉e-mail: pn21@nyu.edu; g.diprofio@itm.cnr.it

present, the most energy-efficient desalination technologies are based on membrane desalination[2,6], with the reverse osmosis (RO) accounting for over 60% of the desalination capacity worldwide[7]. The performance and efficacy of the desalination membranes largely depend on their structure, topology, and surface chemistry[8], which are determined by the method of their fabrication and cannot be modified at will over their operational lifetime. One common drawback of these membranes is their proneness to fouling[9], which results in decreased mass transfer, reduced selectivity, and ultimately, a significant increase in the energy budget and overall operational costs[10,11]. These issues can be partially alleviated by using chemical cleaning, however, the harsh chemicals (strong acids or bases) used are not only detrimental to the membrane's integrity, but the large scale at which these processes are practiced also raise increasing environmental concerns with long-term, unforeseeable effects. Practical solutions to modify the desalination process by resorting to latest technologies that would minimize energy consumption and environmental impact are therefore germane to maintain a stable and sustainable water–energy–environment nexus[12].

Despite the significant progress that has been made with research efforts along this line of pursuit, the design of antifouling membranes with high productivity and rejection ability remains a formidable challenge that requires fundamentally unprecedented approaches, such as for example those that have been proposed for self-cleaning membrane separation[13]. Over the past decade, several advanced membranes have been suggested, aimed at improvement of some functionalities such as tuneable permeability, enhanced selectivity and fouling resistance[14]. Drawing on inspiration from stimuli-responsive cell channels[15], smart gating membranes have been proposed for water treatment[16] with capability for self-modulation of their pore size and/or surface properties by using gradients in temperature[17], light[18], pH[19], magnetic[20] or electric fields[21], and specific ions or molecules[22,23]. Within the hybrid membrane approach, smart gating membranes have been prepared based on coupling of stimuli-responsive materials with traditional porous membranes[24] or by using surface functionalization[25]. Hydrogels have received a particular research attention, since they can be readily processed into switchable membranes with high permeation and ion rejection[26–29].

Among the available dynamic materials that area capable of changing their shape in response to light, temperature, pressure, etc.[30–33], the so-called thermosalient (TS) crystals are a recently established class of dynamic crystalline materials that are capable of sudden expansion or motion under thermal stimulation[34,35], thereby rapidly transforming thermal energy into mechanical work[36–38]. A distinct asset of the TS crystals that is currently not available with any other soft dynamic material is their rapid, efficient and reversible conversion of heat (kinetic energy) into mechanical work at a millisecond scale due to a martensitic phase transition, a property that has been already considered for applications in electronics such as actuation[37], electrical fuses[39], and thermal sensors[40]. These transitions are oftentimes (but not always) preceded by colossal positive or negative expansion of the lattice of the material and, except for a few known cases, they usually end up with a disintegrative outcome. Recently, stabilization by embedding the crystals in soft media such as hydrogels or polymers has been proven to sustain the crystals' integrity, and this opens prospects for maintaining a cyclic operation[41,42]. In this work we capitalize on the dynamic response from TS solids to prepare an innovative class of smart hybrid membranes with gating capability, and we demonstrate that the resulting hybrid membranes are capable of self-cleaning during osmotic and membrane distillation processes. We further report a substantial enhancement in mass transfer by more than 43% for optimized membrane composition and extended operational lifetime at favourable ion rejection ability, properties that qualify these advanced materials as distinct and prospective emerging class of membranes for a variety of separation applications.

## Results and Discussion

To prepare the hybrid membranes, porous polyvinylidene fluoride (PVDF) membranes were combined with a thin layer of polyvinyl alcohol (PVA) hydrogel containing randomly dispersed and oriented TS crystals (for details on the procedure for preparation of the membranes, see the Methods section). As a typical, well-studied and readily affordable TS material, we focused on 1,2,4,5-tetrabromobenzene (TBB), a compound that is known to undergo a TS phase transition only slightly above room temperature and is therefore considered an energy-efficient dynamic crystalline material (Fig. 1A)[43]. TBB is stable at room temperature in its β phase, but undergoes a first-order TS transition to its γ polymorph at 39–46 °C with a vigorous mechanical response (Fig. 1D, E), whereby single crystals suddenly expand and, if they are unrestrained, can even jump several centimeters high as a result of the sudden release of elastic energy that develops in their interior[44–46]. The process of preparation of the membranes is sketched in Fig. 1B. Briefly, aqueous solutions of PVA and glutaraldehyde (GA) are mixed, and different amounts (Fig. 2A) of TBB crystals (insoluble in water) are added to the mixture. The suspension is then treated with hydrochloric acid and immediately and uniformly cast over the porous PVDF membrane support (0.2 μm nominal pore size) by using an automatic film applicator.

Complete polymerization occurred after keeping the films in a hood at room temperature for 24 h. The resulting composite P-P-T membranes were of size >18 cm² and comprised a PVDF support and a PVA hydrogel layer containing dispersed TBB crystals with loading of 0.1–2.0 mg cm⁻² (Fig. 1C, F, H; for the loading, see Fig. 2A). Except for the highest loading of 2.0 mg cm⁻², the membranes had a homogeneous surface and rubbery texture. They were mechanically stable to bending and torsion, and showed good adhesion between the PVDF base and the PVA-TBB coating, even after being stored in water bath or in air at room temperature over 3 months. On the contrary, under identical conditions the membranes having high TBB loading (2.0 mg cm⁻²) were found to delaminate and, after drying, displayed a glassy consistence of the functional layer that easily fractured upon application of mechanical force. These results indicate that there is an upper limit to the loading of the TBB crystals where the membrane retains its compliant mechanical properties, and it becomes brittle at higher TBB concentrations.

To assess the mass transport properties of the P-P-T hybrid membranes in thermostatic conditions, the pure water transmembrane flux (J) was measured in an osmotic distillation (OD) system at different temperatures (Fig. 2B). In the OD process, the aqueous solution that is in contact with the hydrophobic membrane cannot enter the pores in the liquid phase and a liquid–vapour interface is formed at either pore terminus. Since the process is isothermal, the driving force for water evaporation is the difference in chemical activity of the water between the two solutions established due to the different nature and concentration of the solutes[47]. This water activity difference induces water vapour pressure gradient; consequently, the water evaporates from the solution of higher water activity (diluted solution at feed) and the vapor is transported across the membrane pores to condense in the solution of lower water activity (osmotic or drying solution at the distillate side). It is known that the increase of the hydrogel mesh size with temperature has a positive effect on the mass transfer and transmembrane flux (for details, see the Methods section)[48–54], and therefore we anticipated that increasing temperature has a positive effect on the mass transfer and transmembrane flux.

In line with the expectations, J shows an incremental trend with the operating temperature T for the reference membranes without TBB (Fig. 2B). With low TBB loading of 0.5 mg cm⁻² no significant improvement in transport compared to the reference membranes could be observed over the entire temperature range. However, at higher loadings of TBB of 0.8 and 1.0 mg cm⁻² slight enhancement in the flux J was observed in the temperature range 28–40 °C (Fig. 2B). For

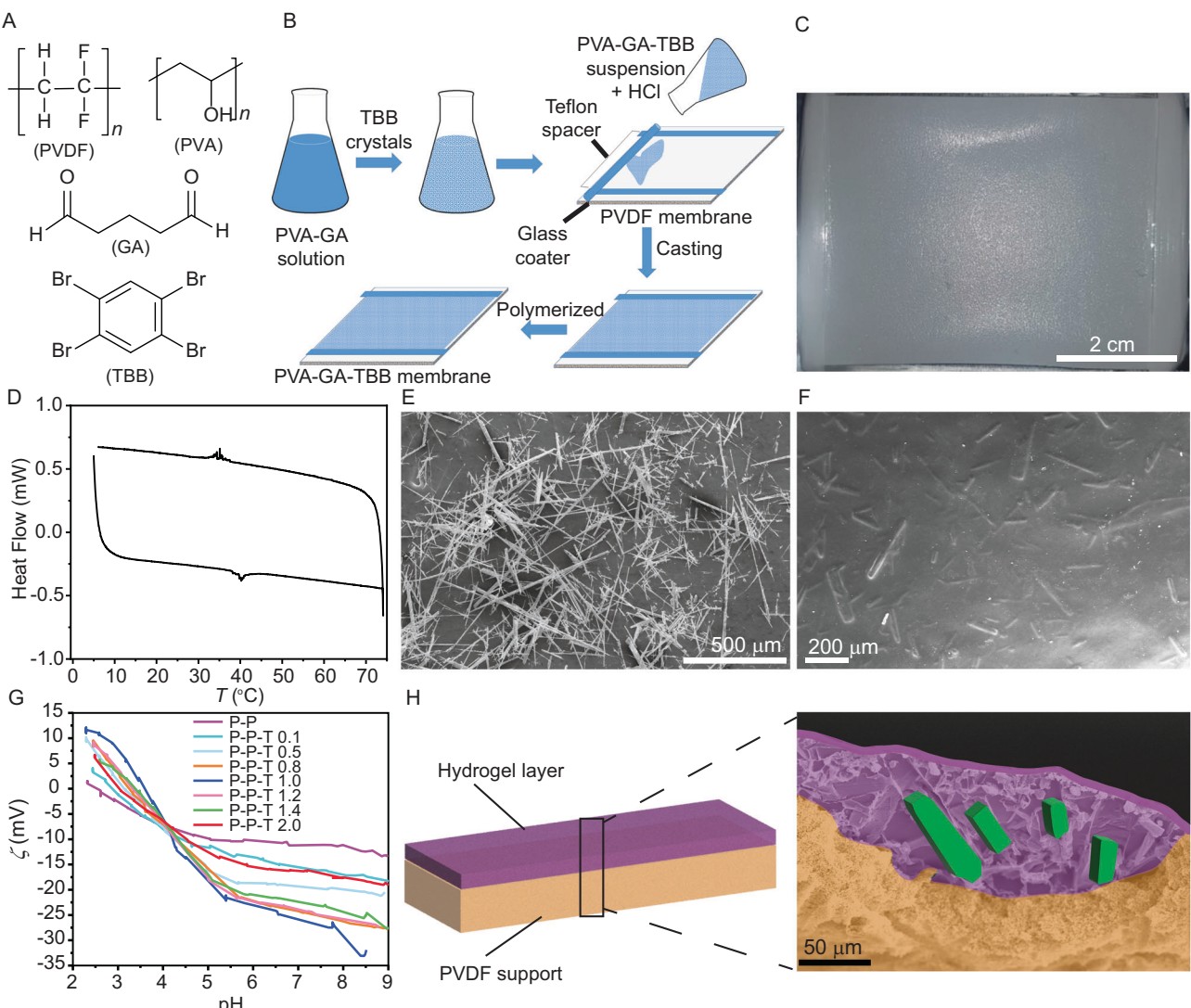

**Fig. 1 | Concept, morphology and characterization of the smart hybrid membranes. A** Chemical structure of the smart membrane components: polyvinylidene fluoride (PVDF), poly(vinyl alcohol) (PVA), glutaraldehyde (GA) and 1,2,4,5-tetra-bromobenzene (TBB). **B** Schematic illustration of the process for preparation of the hybrid membranes. **C** Optical image of one of the membranes (sample P-P-T 1.0). **D** Differential scanning calorimetry (DSC) profile of TBB showing a thermal signature of the reversible TS phase transition at 39–46 °C. **E** Scanning electron microscopy (SEM) image of TBB microcrystals before incorporation. **F** SEM micrograph of TBB crystals embedded in the hydrogel layer of the hybrid membrane (sample P-P-T 0.5). **G** Zeta potential ($\zeta$) of the membranes surface as a function of the pH for composite membranes with different TBB loading (P-P stands for PVDF-PVA and P-P-T for PVDF-PVA-TBB). **H** Schematic cross-section of a composite membrane (sample P-P-T 2.0) along with false-colored SEM image showing the main components of the hybrid membrane in different colors. The membrane composition with the acronyms that refer to the TBB loading are provided in Fig. 2A.

TBB loading of 0.8 and 1.0 mg cm$^{-2}$ at 48 °C, which is beyond the phase transition of TBB, $J$ improved to 1.00 L h$^{-1}$ m$^{-2}$ and 1.15 L h$^{-1}$ m$^{-2}$ respectively, which represents an increase of 24% and 43% relative to 0.8 L h$^{-1}$ m$^{-2}$ for the reference membranes.

As the TBB amount increased further, however, the transmembrane flux decreased for the same temperature, with the values always being comparable to those of the reference membrane for loading of 1.4 mg cm$^{-2}$. Notably, composites with high TBB content, 2.0 mg cm$^{-2}$, had consistently lower flux compared to the reference membranes across the whole operating temperature range, although they also showed a slightly higher flux when taken over the phase transition temperature.

The results in Fig. 2B and the above discussion clearly indicate—from the TBB loadings studied in this work—an optimal crystal loading, 1.00 mg cm$^{-2}$, that appears to maximize the transmembrane flux. They also show that above a certain threshold loading of crystals, there is a second regime for transport of water vapour molecules across the

membrane that is triggered by temperature change across the phase transition region. This observation is perhaps better visualized by the Arrhenius plot in Fig. 2D. The P-P and P-P-T membranes with loading of 0.5 mg cm$^{-2}$ follow the same trend over the entire temperature range, as it is expected for an OD process. The deviation from such trend is evident for TBB loading exceeding 0.5 mg cm$^{-2}$ near the phase transition region. This result indicates that, above a threshold concentration of the additive dispersed in the functional layer (approximately 0.5 mg cm$^{-2}$) and across the transition region of TBB a different transport mechanism across the membrane is activated, likely due to the change in the physical state of the dynamic crystals in response to heating. Since in all experiments the rejection factor ($R$%) to NaCl exceeded 99.95% (Supplementary Table 1), this result also indicates that the water mass transport across the composite membranes continues to occur in the vapor phase, so that there is no mixing between the feed and the drying solution in liquid state (no membrane wetting) as a consequence of the different transport mechanism under heating.

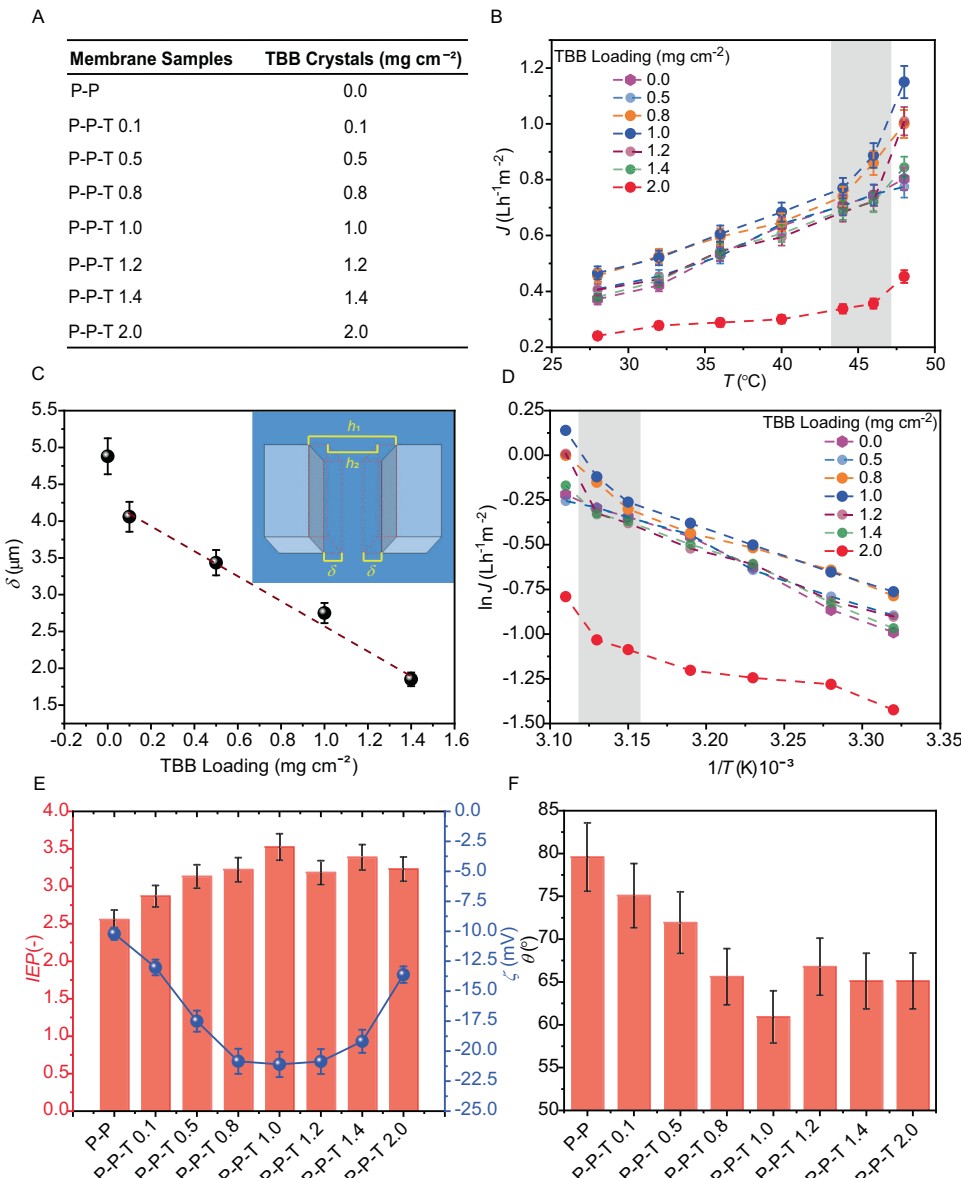

**Fig. 2 | Characterization of the transportat properties of hybrid membranes with different TBB loading. A** Sample codes used throughout the text and in the figures referring to the different loading of TBB crystals in membranes with different compositions. P-P stands for PVDF-PVA and P-P-T for PVDF-PVA-TBB. **B** Temperature dependence of the pure water transmembrane flux $J$ of the hybrid membranes with different TBB loading. The temperature range where the phase transition of TBB occurs is shaded. **C** Deformation of the PVA hydrogel layer ($\delta$) in the direction perpendicular to the membrane surface as a function of the TBB loading. Inset: schematic definition of PVA layer displacement upon swelling. The plot is a linear fit including the last four data points (the first one was excluded from the fitting). **D** Arrhenius plot of the transmembrane flux against the reciprocal absolute temperature for composite membranes with different TBB loading. The temperature range of the transition temperature region for TBB is shaded. **E** Isoelectric point (*IEP*) and zeta potential ($\zeta$) as a function of TBB loading in the composite membranes. **F** Water contact angle ($\theta$) measured at the surface of membranes with different TBB loading in the hydrogel layer. The acronyms in panels (**E**) and (**F**) correspond to the loading shown in Fig. 2A. The error bars in Fig. 2B, C, E, F represent standard deviations over three measurements.

The effect of the TBB loading on the mechanical properties of the hydrogel layer—a performance aspect that could be relevant in view of the mechanical robustness of the membranes for implementation in a desalination setup—is outlined in Fig. 2C, which shows uniaxial displacement of the gel upon hydration and swelling in the direction perpendicular to the membrane surface as a function of the amount of TBB. Inclusion of crystals in the native (undoped) membrane, even at concentration as low as 0.1 mg cm$^{-2}$ significantly stiffens the membrane, as seen with the drop in membrane displacement. Further increase in TBB loading results in nearly linear decrease in mechanical deformation. We hypothesize that the inclusion of the small needle-like crystals of TBB within the hydrogel and their entanglement with the net-like structure of the hydrogel hinders the ability of the coating

to undergo elastic deformation. Partial crystal interdigitation on the top of the surface probably also contributes to the apparent change of its consistence from rubbery to glassy, which was particularly evident with the loading of 2.0 mg cm$^{-2}$. In this latter case, the membrane was sufficiently stiff to prevent us from measurement of its mechanical deformation due to immediate delamination that occurred between the PVDF and hydrogel layers as a result of the significant difference in their individual stiffnesses.

Aimed to investigate the reason for the effect of TBB crystals on the transmembrane flux, the surface of the composite membranes was also characterized by measurement of the isoelectric point (*IEP*), zeta potential ($\zeta$) and water contact angle ($\theta$). Figure 2E shows that while the isoelectric point (the charge that develops at the interface between the

membrane surface and the liquid medium) increases with the amount of TBB crystals dispersed in the hydrogel layer up to 1.00 mg cm$^{-2}$, and there is a slight decrease at higher loading. The $\zeta$ potential measured at the pH of the feed solution (5.35) shows the opposite trend: the surface initially becomes more negatively charged as the amount of TBB increases, and increases above 1.00 mg cm$^{-2}$. This observation is due to the crossing of the $\zeta$ vs. pH curves at pH ~4.3, as shown in Fig. 1G, a value that corresponds to the p$K_a$ of the acetate groups that remain in the partially hydrolyzed PVA. Namely, PVA is usually prepared by hydrolysis of poly(vinyl acetate), a process that is incomplete in alkaline medium[55], resulting in an atactic copolymer poly(vinyl alcohol-co-vinyl acetate). Accordingly, the ATR-FTIR spectra of the composite membranes (Supplementary Fig. 4) contain major peaks associated with partially hydrolyzed and cross-linked PVA[56–59]. The peak around 3330 cm$^{-1}$ is due to the stretching vibration of hydrogen-bonded hydroxyl groups (O−H) and corresponds to strong inter- and intra-molecular hydrogen bonds among the PVA chains. The C−H stretching modes of CH$_2$ of PVA backbone appear at 2917 cm$^{-1}$ and 2865 cm$^{-1}$. The C–O stretching band was observed at 1092 cm$^{-1}$, whereas the band at 995 cm$^{-1}$ can be attributed to the bridges (–C–O–C–) obtained by reaction between the hydroxyl groups of PVA and the cross-linker, glutaraldehyde. The characteristic C=O stretching from the poly(vinyl acetate) and the acetate groups is retained in the partially hydrolyzed PVA, and its IR signature was observed around 1650 cm$^{-1}$.

The presence of large hydrophobic acetate groups in partially hydrolyzed PVA weakens both inter- and intra-molecular hydrogen bonding between the hydrophilic hydroxyl groups[60–62]. When the pH is raised from 2.5 to 6, the OH$^-$ ions hydrolyze the remaining acetate groups. The polymer chains then become highly negative; as these like-charges repel each other, the swelling ratio increases[63]. By further increase of pH from 6 to 9, the diffusion rate decreases, owing to the enhanced ionic strength that is expected to neutralize the negative charges of the acetate groups[64]. Thus, the charge carried by the PVA at high degree of hydrolysis used in this work (> 99%) is affected by the hydrogen bonds and the attractive/repulsive electrostatic interactions among the functional groups. The presence of TBB crystals in the hydrogel layer appears to be reinforcing the effect of the acetate groups by providing interaction among the OH$^-$ groups and TBB. This effect increases with TBB loading up to around 1 mg cm$^{-2}$, as it can be inferred from the higher slope of the $\zeta$-potential curves that intersect at the pH corresponding to the effective p$K_a$ of the acetate groups (reduced from 4.75 for acetic acid to around 4.3 for the acetate groups in PVA). Across the p$K_a$ range and under the effect of TBB, PVA seems to act as an amphiphilic polymer, where the balance among the hydrophilic and hydrophobic groups in aqueous solution provides sensitivity to pH[60–62]. It follows that the responsivity of the surface charge to the pH of the hydrogel layer in composite membranes (a pH-responsive behavior) is expected to be more pronounced at low degree of hydrolysis of the starting PVA (higher relative amount of acetate groups compared to hydroxyl ions) and at higher TBB loading in the gel layer. However, when the TBB loading is increased above 1.0 mg cm$^{-2}$ the crystals aggregate and their stratification reduces the TBB-to-PVA interaction, thereby decreasing the charge of the hydrogel layer. Generally, we conclude that in all the cases, the presence of TBB crystals clearly affects the mechanical properties of the gel and reduces its propensity for swelling (Fig. 2C), even in presence of the charge effect described above.

The water contact angle ($\theta$) of the P-P samples is about 80° and is reduced with the loading of TBB crystals of down to 60° at 1.00 mg cm$^{-2}$ TBB (Fig. 2F). There is a slight increase of $\theta$ for loading > 1.0 mg cm$^{-2}$, where it remains around 65°. Based on these results, we hypothesize that the presence of TBB crystals in the gel layer of the hybrid membrane increases the water affinity (hydrophilicity), thereby allowing enhanced transport of pure water. On the other hand, the tendency for interdigitation or stratification of the TBB crystals affects

the mechanical properties and turns the composite from elastic to stiff, and reduces both its mechanical stability *and* transport properties. We conclude that, at least from the loadings used here, the TBB loading of 1.00 mg cm$^{-2}$ represents a trade-off between improvement in the transport properties due to enhanced hydrophilicity and reduction of the transmembrane flux caused by crystal aggregation. Thus, the sample P-P-T 1.0 was selected for further studies.

Figure 3A shows the evolution over time of the transmembrane flux in OD when changing the temperature of the system across the transition temperature for a reference (unloaded) and 1.0 mg cm$^{-2}$-loaded membranes with feed solutions containing bovine serum albumin (BSA), humic acid (HA), and sodium alginate (SA) as model membrane foulants. Before the measurement, the temperature was set to 28 °C from ambient temperature (~18 °C). For both membranes, the transmembrane flux increases as the system equilibrates at the higher temperature, according to Eq. 4 in the Methods section. After 2.3 h, the temperature was set to 48 °C. Following a delay where the flux continues to approach towards a plateau, both membranes show an increase in transport, as expected for a higher temperature. We note that the hybrid membrane displays faster response (minor delay) compared to the unloaded one in its adaptation of the flux to the change in temperature, demonstrating the dynamic effect that the TBB crystals exert in driving the mass transport when the system is taken over the phase transition. At 5.3 h, the temperature was set to 28 °C again. After a certain delay (1 h), the flux at the reference membrane starts to decrease. After more than 2 h since the last temperature switch, the hybrid membrane with 1.0 mg cm$^{-2}$ TBB continued to show an increasing flux, with a value at the end of the experiment (7.7 h) that was more than 45% higher than that of the reference sample (7.2 × 10$^{-1}$ L h$^{-1}$ m$^{-2}$, against 4.9 × 10$^{-1}$ L h$^{-1}$ m$^{-2}$).

In order to obtain a deeper insight into the mass transport mechanism that governs the permeation across hybrid P-P-T membranes, gas permeation measurements were performed to arrive at a qualitative evaluation of the mean pore size of the membranes (Fig. 3B–F; for experimental details, see the Methods section). Generally, the transport mechanisms depend directly on the structure, nature, and morphology of the membrane[54]. Membranes suitable for gas separation can be categorized into two broad families−porous and non-porous. In the case of porous membranes, viscous flow, Knudsen diffusion and molecular sieving are usually the underlying factors of their transport mechanisms[54]. Supplementary Figure 3 shows two examples of typical membrane pores with different diameter and a number of gas molecules permeating through the membrane. The viscous flow normally occurs in the pores with a diameter larger than 50 nm, where the energy loss during transportation is mainly due to interactions between the molecules. In the pores with a diameter between 2 and 50 nm, the transport of molecules through the pores is similar to the diffusion in a homogeneous phase, where the interaction of molecules with the pore wall is the main factor responsible for the energy loss during the permeation. This Knudsen flow transport mechanism typically occurs when the pores have a diameter that is smaller than the mean free path length of the diffusing gas molecules[65]. In porous structures, an increase in temperature typically reduces the permeation of gases. Specifically, when the transport is dominated by the Knudsen mechanism, the gas flux is inversely proportional to the square root of the temperature (Eqs. 3 and 4 in the Supplementary Methods). When the transport is controlled by a viscous flow, the negative effect of temperature is due to an increase in gas viscosity.

The data on gas permeability as a function of temperature reported in Fig. 3B−F shows that in all cases of the membranes studied here H$_2$ has the highest permeability. It is followed by N$_2$ and CO$_2$ for blank and a membrane loaded with 0.1 mg cm$^{-2}$ TBB, whereas CO$_2$ is more permeable than N$_2$ for membranes having higher load. Overall, the presence of thermoresponsive TBB crystals had a small effect on the transport of gases. At 35 °C, the CO$_2$ permeability for the

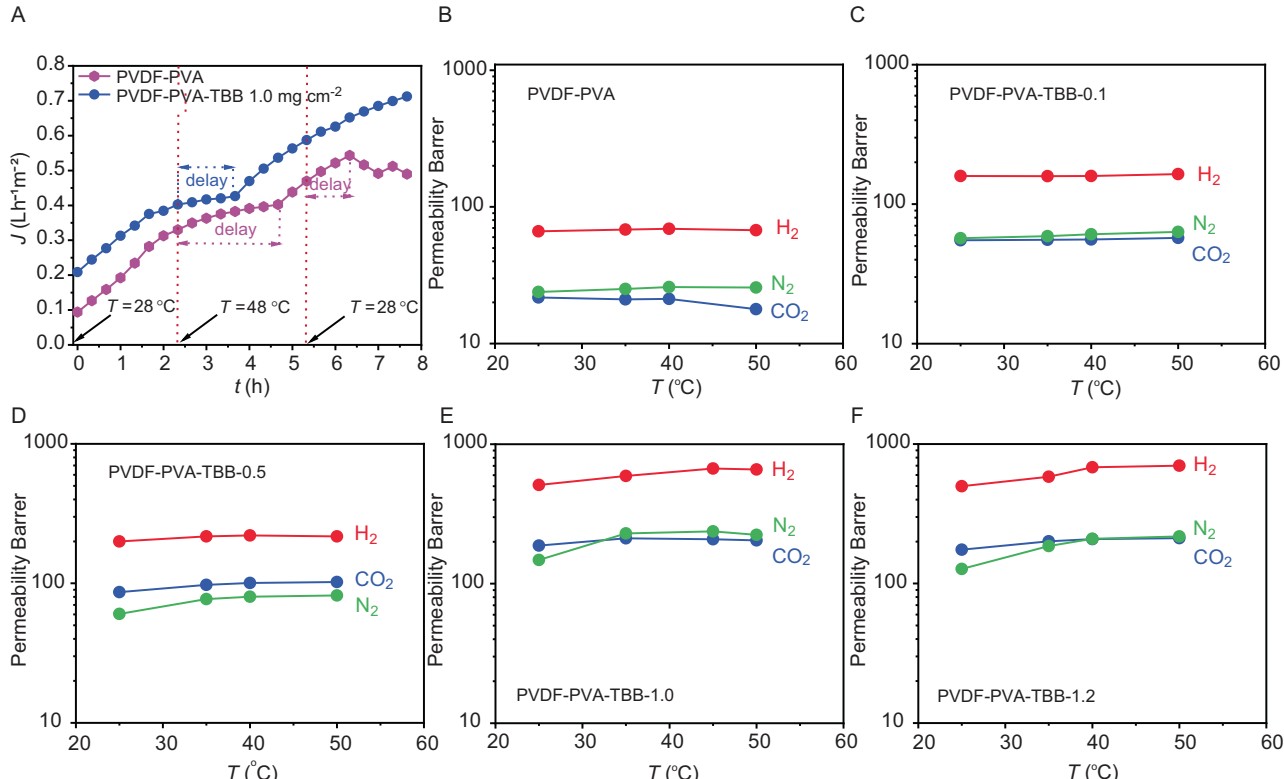

**Fig. 3 | Performance of the smart hybrid membranes in descaling and their gas permeability. A** Simultaneous transmembrane flux for a reference (unloaded) membrane and the 1.0 mg cm$^{-2}$ TBB-loaded membrane with solution containing model foulants (BSA, HA and SA), when changing the temperature of the system across the phase transition temperature. **B–F** Permeability of $CO_2$, $H_2$, and $N_2$ as a function of temperature for membranes at different loading of thermoresponsive particles. The loading is given on each panel, and the sample labels refer to Fig. 2A.

membrane with 1.00 mg cm$^{-2}$ TBB was 12% higher than that measured at 25 °C, against only 4% of the reference. At 50 °C the $CO_2$ permeability was 9% greater against the 11.7% of the reference. The Knudsen selectivities for $H_2/N_2$ and $H_2/CO_2$ gas combinations were found to be 3.7 and 4.7, respectively. The ideal selectivities measured on the tested P-P-T membranes did not vary significantly with the TBB particles loading, ranging from 2.8 to 3.9 and from 2.2 to 3.8 for $H_2/N_2$ and $H_2/CO_2$ gases, respectively, confirming that gas transport through these membranes is mainly governed by the Knudsen mechanism, with some contribution from viscous flow, which explains the lower selectivities compared to the ideal Knudsen values (Supplementary Table 2)[66].

This result confirms the presence of the porous structure in the selective layer of all hybrid membranes, with most of the pores having a diameter between 2 and 50 nm. For this reason, while gas selectivity effects were masked by the presence of pores which have a dominating contribution to the transport through the membrane, the observed increase in gas permeability with temperature in P-P-T membranes (compared to reference membranes) can be attributed to the dynamic effect of TBB crystals responding to heating, which counteracts the typical negative temperature dependence of Knudsen and viscous flow mechanisms. Accordingly, the deviation from an Arrhenius-type mass transport curve across the transition region of Fig. 2D is clearly due to the dynamic effect of the TBB crystals in response to heating.

The decline in average transmembrane flux with the undoped membrane in successive OD water desalination cycles with foulant molecules (BSA, HA, and SA) is shown in Fig. 4A. After five cycles of reuse for the same membrane, the transmembrane flux decreases by more than 40%, from 6.3 to 3.7 × 10$^{-1}$ L h$^{-1}$ m$^{-2}$ due to steady accumulation of foulants on the surface. The foulants are not removed by simple flushing of the membrane with pure water and affect the transport performances in the following cycle. On the other hand, for doped P-P-T 1.0 membrane, apparent is a slight increase in the average

value of $J$ on going from the 1st to the 4th cycle, with substantial decline only in the 5th reuse. Nevertheless, after five cycles, the average flux for TBB-loaded membranes is still higher than 160% of that of the undoped sample. While these results clearly demonstrate the antifouling behavior of P-P-T membranes compared to their undoped counterparts in the OD process, the results show the strong effect of TBB on the transport properties as the same membrane is subjected to successive heating and cooling cycles from 48 to 22 °C. Only at the 5th reuse this effect appears to be alleviated, and the fouling starts to affect the membrane performance.

For both undoped and TBB-loaded membranes, it can be concluded from Fig. 4B that the instantaneous transmembrane flux shows an increasing asymptotic trend with the operating time in multiple cycles of operation (Supplementary Figs. 5, 6). This is an unexpected behavior, since as the feed solution concentrates because of solvent removal in the vapor phase, and the drying solution dilutes due to pure water condensation, the driving force of the process (the gradient in vapor pressure between the two membrane sides) progressively decreases with time and affects the transport properties. Inspection of the surface by using SEM (Fig. 4C–F) shows that the dense surface structure of the functional layer develops a porosity during the heating and cooling cycles. This is an intrinsic effect of the PVA layer, and it was observed for both unloaded (Fig. 4C, D) and TBB-loaded membranes (Fig. 4E, F). Additionally, large irregular pores or holes were also observed with the doped membranes, and are attributed to popping out of some TBB crystals during the phase transition (Fig. 4F). Such motions, accompanied by partial disintegration, are expected for the TS effect of the crystals. Therefore, upon heating, the composite membrane facilitates an increase in the instantaneous transmembrane flux with time, until an equilibrium close to an upper limit has been reached.

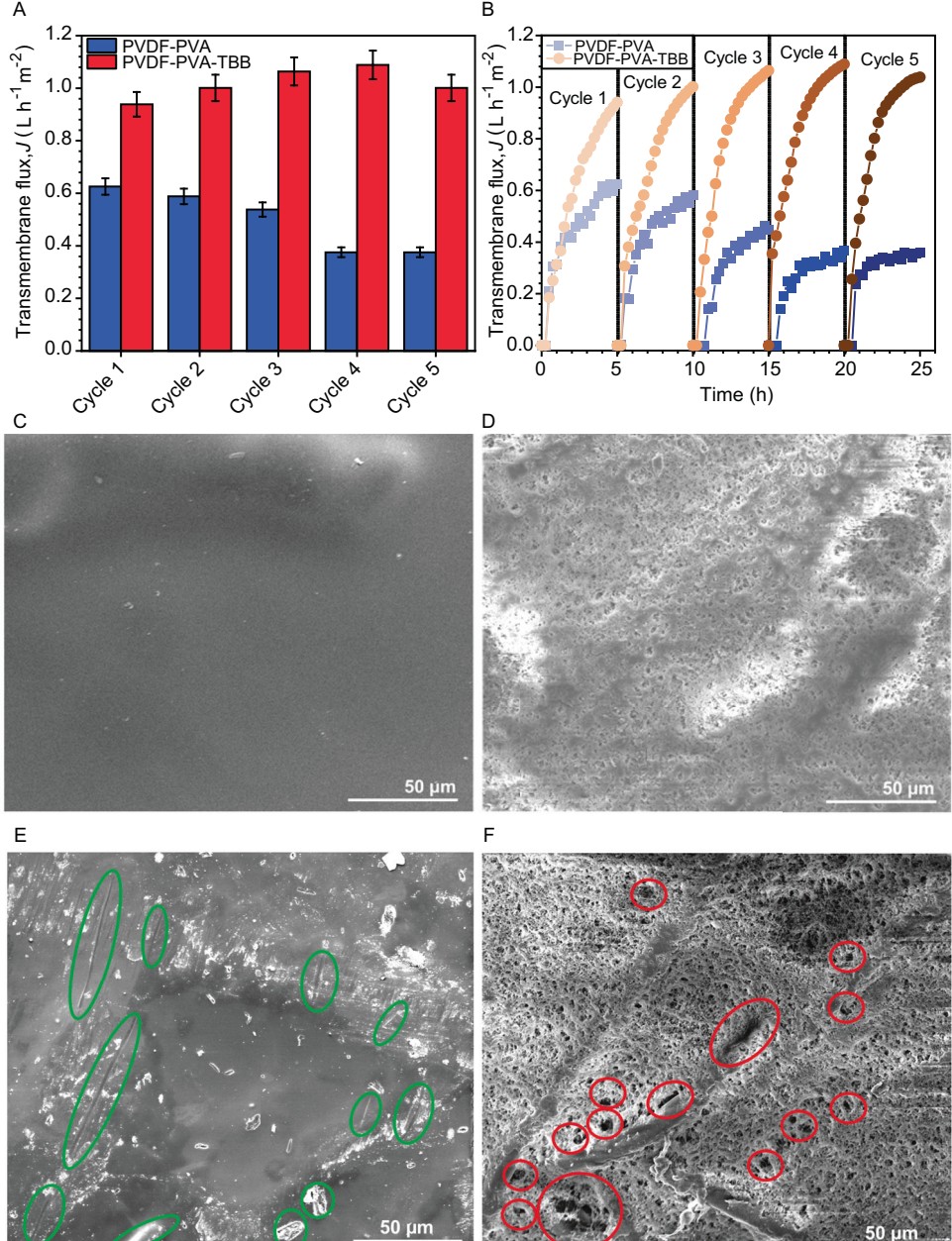

**Fig. 4 | Cycling performance of the membranes and surface morphology.**
**A** Total transmembrane flux for unloaded PVDF-PVA and PVDF-PVA-TBB
1.0 mg cm⁻² membranes in multi-cycle OD tests carried out at 48 °C calculated at
the end of each cycle. The error bars represent standard deviations for three
measurements. **B** Instantaneous transmembrane flux as a function of the operating
time for undoped and PVDF-PVA-TBB (doped) membranes in multiple OD cycles.
After each cycle, the membranes were flushed with milliQ water for 20 min at room
temperature. **C**, **D** SEM micrographs of undoped PVDF-PVA membrane at room
temperature (**C**) and after heating to 50 °C (**D**). **E**, **F** SEM micrographs of doped
membrane PVDF-PVA-TBB 1.0 at room temperature (**E**) and after heating to 50 °C
(**F**). The green circles indicate the embedded TBB crystals while the red circles
indicate the large pores or holes generated after some of the TBB crystals have
detached during the phase transition.

In case of the P-P-T membranes, exponential fittings of the experimental data in Fig. 4B (see Supplementary Methods) indicate that the limiting value $y_0$ remains around 1.1 L h⁻¹ m⁻², irrespective of the membrane reuse (Supplementary Table 3). Furthermore, the curves in Fig. 4B appear to become steeper with the membrane reuse. This observation is further supported by the fitting parameter $t_1$ in Supplementary Table 3, which reflects the time necessary to reach the limiting value, whose absolute value decreases regularly from the 1st to the 5th cycle. On the other hand, the limiting values for undoped samples decrease regularly from 0.62 to 0.36 L h⁻¹ m⁻² from the 1st to the 5th cycle. Moreover, an earlier flattening of the curve with the progressive reuse can be seen from Fig. 4B (as well as from the $t_1$ values

in Supplementary Table 3), although that is less evident than for the TBB-loaded samples, and comes with larger data scatter due to decreased mechanical stability. When both samples are treated in successive thermal cycles, their porosity is enhanced sooner, and therefore they reach the limiting transmembrane flux earlier. This effect seems to be reversible to some extent, perhaps as a result of self-healing mechanism(s), since at the beginning of each cycle the instantaneous transmembrane flux increases progressively starting from values close to zero. Based on this analysis, it becomes clear that for undoped membranes, in addition to the consistent reduction of the limiting flux relative to P-P-T in the respective cycles, the increased surface porosity under thermal gradient does not allow for mitigation

of the degradation of the transport properties due to fouling, and therefore the asymptotic flux decreases with membrane reuse. Conversely, TBB-loaded membranes are unaffected by the foulants of the feed solution because the limiting value of $J$ is independent from the membrane reuse within the range of investigated conditions, while the time necessary to reach this limiting flux is a function of the cycle number.

In order to illustrate the antifouling action in thermoresponsive membranes, various mechanisms have been discussed in the literature based on "open-close" gating valves or pores[26,67], polymer shrinking–stretching effect, or formation of a buffer layer that eliminates the attraction force between pollutants and membrane[68]. In contrast with these examples, our proposed mechanism for the smart hybrid membranes described here is based on thermo-responsive capability to control the flux and fouling aptitude based on mechanical actuation during a phase transition of organic crystalline material. Indeed, pore formation after heating that we observe with these membranes could contribute to the increase in flux under thermal treatment. However, since we observed evolution of additional, larger pores or holes for doped membranes, we conclude that this effect could also be responsible for the increase in flux compared to undoped samples. By comparing the performance of the composite membranes with the reported thermoresponsive membranes, we find a similarity with the reported work on microgel based thermoresponsive membranes which can also change their performance when subjected to a thermal gradient[26,27], although the reported membranes show higher water flux. Moreover, Lyly et al. reported thermal cleaning efficiency of thermoresponsive membranes against model organic foulants such as BSA and SA in conjunction with thermal actuation[28,29], a concept that is closest to our work since, although we use TS TBB crystals in a composite membrane to instigate thermal cleaning efficiency. Although, to the best of our knowledge, there are no prior reports on membranes based on the TS effect that would be compared directly to the materials reported here, in Supplementary Table 4 we compiled a comprehensive and exhaustive list of separation membranes. This database not only provides a basis for benchmarking and assessment, but also aids in the explanation of the difference in operating conditions of the advanced family membranes that we report here. This comparison highlights the distinct properties of the smart materials reported here and those reported earlier.

The best-performing membrane composition in OD (P-P-T 1.0) was tested in consecutive cycles of direct contact membrane distillation (DCMD) experiments using hypersaline feed solution containing solute salts at 228 g L$^{-1}$ and foulant molecules (Supplementary Table 5, Supplementary Figs. 7, 8). The feed was warmed up to 59 °C and thus beyond the transition temperature of TBB, and the condensing water at the distillate side was set to 20 °C. Figure 5A depicts the average transmembrane flux measured for both TBB-loaded and nascent membranes during the consecutive DCMD cycles using the same membrane, with intermediate flushing of 15 minutes with distilled water preheated to 40 °C. Despite the small driving force used during the process ($\Delta T = 39$ °C) and the poor fluid-dynamic conditions (Reynold number $Re = 24$, assuming a solution density of 1151.5 kg m$^{-3}$ at 59 °C) that have minor effect on heat and mass transfer towards the membrane boundary layer[69], in all cycles $J$ was found to be over 1 L h$^{-1}$ m$^{-2}$ for TBB-doped membrane, with a decrease of about 7% from the first to the third cycle. At such high salinity of the feed, which is well beyond the range of operability of reverse osmosis for water desalination due to osmotic pressure limitations[70], the thermal energy consumption with P-P-T 1.0 membrane was around 9500 kWh m$^{-3}$ and it remained nearly constant for several cycles (Fig. 5B). As comparison, for undoped P-P membrane, the decrease in $J$ was as high as 40% while the thermal energy consumption increased by almost 60% from the first to the third cycle.

However, since the flux depends on the applied driving force, the overall membrane mass transfer coefficient $B_m$ was calculated[69] to enable a more direct comparison between TBB-doped and undoped membranes. Indeed, $B_m$ allows evaluation of the real permeability performance of the membrane, apart from the effect of the driving force[71]. Fig. 5A shows a steeper decrease in the mass transfer coefficient with membrane reuse for undoped membranes, which means that there is an increased mass transfer resistance due to accumulation of foulants or decrease in membrane permeability that is less effective for TBB-loaded samples. This data clearly demonstrates that the transmembrane flux is directly related to the intrinsic mass transfer properties of the membrane itself, independent of the effect of the driving force which is identical across all tests. Furthermore, as the membrane is reused over multiple cycles, the mass transfer properties of the P-P-T 1.0 membrane indicate that it is less prone to fouling, demonstrating the self-cleaning properties of the TBB-loaded membranes.

In order to assess the effect of thermal expansion of TBB at temperatures below the phase transition, we analyzed the structure of TBB by using variable-temperature single crystal X-ray diffraction. A slight shift in the diffraction peak position was observed between 275 and 310 K before the phase transition, consistent with thermal expansion of the crystal lattice. The thermal expansion coefficients were determined by diffraction between 275 and 310 K[72]. Before the phase transition, the TBB crystals undergo negative thermal expansion along the principal X$_1$ axis [0.2673, −0.0, 0.9636] with coefficient of −188.5373 MK$^{-1}$ and along the X$_2$ axis [−0.0, 1.0, −0.0] with −252.5461 MK$^{-1}$. This is compensated with a positive thermal expansion along the X$_3$ axis [0.9995, −0.0, 0.0311] with a coefficient of 234.3019 MK$^{-1}$ (Supplementary Fig. 12, Supplementary Tables 6–9). The strongly anisotropic thermal expansion is expected to affect the contact between the crystals and the matrix, and could lead to enhanced transport properties of the membrane. Additional experiments were carried out to study the dielectric properties of pristine and doped membranes using electrochemical techniques which are widely used to characterize the polarized membrane behavior[73–75]. It was observed that the doped membrane has 25% higher dielectric constant ($\varepsilon = 2.74 \pm 0.04$) compared to the pristine membrane ($\varepsilon = 2.06 \pm 0.06$) (Supplementary Fig. 9).

In summary, we described a conceptually unprecedented approach to smart gating membranes where dynamic functionality is added to otherwise static membranes by surface coating with polymer-embedded TS crystals. We demonstrated that the smart membranes are capable of gating, whereby the mechanical instability of the composite, driven by the dynamic response of the crystals to heating slightly above room temperature, allows a pure water flux increase exceeding 40% for optimal TBB loading conditions in saltwater desalination by OD, compared to reference membranes. Furthermore, the inclusion of TBB crystals in the crosslinked PVA network provides an added value of enhanced fouling resistance, imparted by the enhanced hydrophilicity and the phase transition property of the dynamic crystals in both OD and MD. Since the surface charge and hydrophilic character of the functional coating is tuneable with the pH of the solution, the proposed approach allows a double responsive mechanism—by pH and temperature—to increase the transmembrane flux, remove foulants, and extend the membrane operational lifetime in water desalination by DCMD under conditions that are well beyond the salinity operability range of reverse osmosis. Within a broader context, considering the existing possibility to recover the enthalpy of water vaporization by distillate condensation in multiple stages to reduce thermal energy consumption, and the availability of more than twenty TS crystalline materials and different membrane compositions, this proposed approach is the first step towards the development of a great variety of smart, energy-conserving, hybrid separation membranes. This advanced family of membranes are endowed with

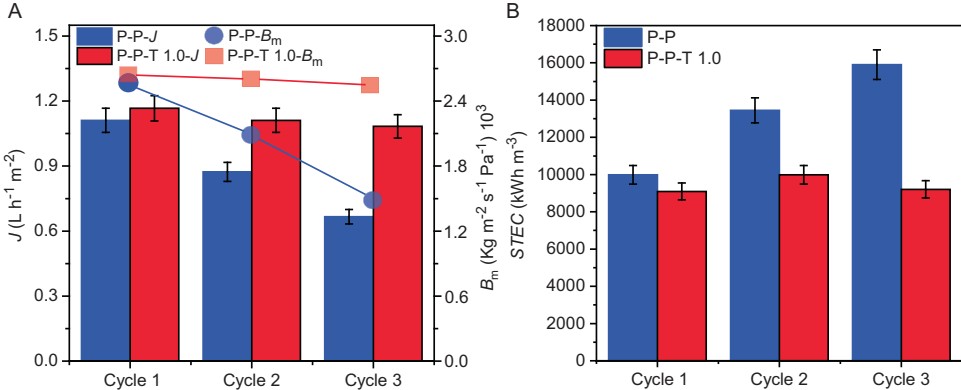

**Fig. 5 | Multiple cycling DCMD tests of the membranes. A** Transmembrane flux ($J$) and overall mass transfer coefficient ($B_m$) for unloaded PVDF-PVA (P-P) and PVDF-PVA-TBB (P-P-T) membranes, the latter loaded with 1.0 mg cm$^{-2}$ (P-P-T 1.0), in multiple cycles of DCMD tests with hypersaline feed solutions at 228 g L$^{-1}$ TDS and foulant molecules as feed. The membranes were flushed for 15 min with preheated water at ~40 °C between two consecutive runs. **B** Specific thermal energy consumption (STEC) for the DCMD tests. The error bars represent standard deviations over three measurements.

capability to respond to various stimuli while they could also alleviate or completely overcome the fouling problem simply by short-term exposure to the respective stimulus.

## Methods
### Materials and basic characterization
Poly(vinyl alcohol) (PVA; MW 85,000–146,000 g mol$^{-1}$, > 99% hydrolyzed, ID: #363146, Sigma-Aldrich) and glutaraldehyde (GA, 25% in water, Merck) were used as monomer and crosslinker, respectively. Hydrochloric acid (HCl, 37%, AR grade, ID #433160, VWR) was used as acid catalyst. Microcrystals of 1,2,4,5-tetrabromobenzene (TBB, 97%, Sigma-Aldrich) were prepared from a saturated solution in tetrahydrofuran (THF) at room temperature after several filtrations. The microcrystals were filtered as soon as they appeared at the bottom of the flask after cooling of the hot solution. Differential scanning calorimetry (DSC) was performed on a TA DSC-Q2000 instrument at the rate of 5 °C min$^{-1}$. QUANTA FEG 450 electron microscope (Thermo Scientific) was used to record the scanning electron micrographs. For the DCMD and OD tests, NaCl (ID #27810.295, VWR), and MgCl$_2$·6H$_2$O (ID #105833, Merck) were used to prepare feed and drying (osmotic) solutions, respectively. For the fouling tests, lyophilized powder of bovine serum albumin (BSA, MW ~66 kDa, assay ~99%, Sigma-Aldrich), sodium alginate (SA, ID #36655, Carlo Erba) and humic acid (HA, ID #GA11745, Fluka) were added to the feed solution.

### Preparation of the PVDF-PVA-TBB membranes
The PVA polymer powder (5% w/v) was dissolved in distilled water by heating (~80 °C) and stirred until a clear homogeneous solution was obtained. Aqueous solution of PVA was mixed with a solution of glutaraldehyde (0.5% v/v) as a crosslinker. Different amounts of the insoluble TBB crystals were added to the mixture of PVA of GA. The amount of added TBB determines the membrane area to be coated in order to obtain loading from 0.1 to 2.0 mg cm$^{-2}$ (Fig. 2A, Supplementary Fig. 13). The suspension of PVA, GA and TBB was subjected to gentle stirring immediately before the crosslinking reaction to ensure good dispersion of the crystals.

The PVDF membrane was washed with water for 1 h at room temperature, dried and fixed on a glass support by Teflon tape used as a spacer to obtain the desired thickness of 400 µm. Before the coating step, hydrochloric acid (37%), was added as a catalyst to the P-P-T suspension. Then the prepared pre-polymeric suspension was homogeneously cast on the top of the PVDF membrane surface using a film applicator (Elcometer Limited, UK) and left to polymerize in a hood for 24 h at room temperature. P-P composite membranes without TBB

crystals were also prepared and used as a reference. Before use, the composite membranes were extensively washed with distilled water to remove the unreacted polymer. Both the reference and the TBB-dispersed membranes were left in water bath and air, at room temperature, for three months to test the mechanical stability and the adhesion between the hydrogel layer and the polymer support.

### Characterization of the composite membranes
The composite membranes were inspected by using an optical microscope (Nikon Eclipse LV100ND) equipped with a video-camera. The contact angle to water ($\theta$) of the composite and reference membranes was measured using CAM 200 contact angle meter (KSV Instruments Ltd.) equipped with a microsyringe and an automatic dispenser at room temperature. A small drop of 5 µL of water was placed on the sample surface and the contact was measured immediately. Five sequential measurements were averaged for each membranes sample. The surface charge (zeta potential, $\zeta$) and the isoelectric point (*IEP*) of the membranes were measured by a Surpass3 (Anton Paar) zeta potential analyser. The zeta potentials were measured using 0.005 mol L$^{-1}$ KCl aqueous solution with pH 2 – 9 at room temperature.

The uniaxial displacement ability $\delta$ of the composites due to swelling were evaluated by the zeta potential measurement device. In the measuring cell, a defined distance (gap height) between the two opposite membrane surfaces was initially set to $h_1$. During the measurement, the electrolyte solution flowed through this gap and caused swelling of the thin film of PVA hydrogel in the direction perpendicular to the surface, allowing the gap height to decrease to $h_2$. The dynamic displacement of the thin hydrogel layer was calculated as:

$$\delta = \frac{1}{2}(h_1 - h_2) \tag{1}$$

Scanning electron microscope (SEM; EVO MA10 Zeiss & Quanta FEG 450) was employed to examine the morphology of the membrane surface and their cross-sections. To inspect the surface, a small piece of a membrane sample was fixed with carbon conductive double side tape to stubs, while for inspection of the cross-sections the membranes were cryo-fractured with liquid nitrogen. All samples were sputtered with a double thin layer of gold under an argon atmosphere for conductivity. Attenuated total reflectance Fourier transform infrared (ATR-FTIR) spectra were recorded in transmittance mode on P-P and P-P-T samples by using a Perkin-Elmer (Massachusetts, USA) spectrometer, in the range of 4000 to 650 cm$^{-1}$. The gas permeation

properties of the composite membranes were measured in an experimental gas separation setup (Supplementary Fig. 1) by using the pressure drop method, while feeding single gas ($CO_2$, 99.995% $N_2$, 99.9995% $H_2$, 99.999%) in dry conditions at different temperatures (25, 35, 40, 50 °C) and operating transmembrane pressure differences (2–8 bar).

## Dependence of the mesh size on the temperature

Assuming absence of boundary layer effects, the water flux across the stagnant film of air within the pores can be generally approximated by Eq. 2[48]:

$$J = B_m P^* (a_1 - a_2) \qquad (2)$$

where $B_m$ is the membrane mass transfer coefficient (assumed as a constant), $P^*$ is the pure water vapour pressure, and $a_1$ and $a_2$ are the bulk water activity at the feed and at the distillate side, respectively. The variation of pure water vapour pressure with temperature is described by the Clausius–Clapeyron equation[49]

$$\frac{dP^*}{dT} = \frac{P^* \Delta H_v}{RT^2} \qquad (3)$$

where $T$ is the absolute temperature, $R$ is the gas constant and $\Delta H_v$ is the water latent heat of vaporisation. By integrating Eq. 3, considering that $\Delta H_v$ and $R$ are independent of the temperature, and by substituting $P^*$ in Eq. 2, an Arrhenius-type dependence with temperature for the water flux is obtained[50]

$$J = J_w^0 \exp\left(-\frac{\Delta H_v}{RT}\right) \qquad (4)$$

where $J_w^0 = A k_{mp}(a_1 - a_2)$, with $A$ being an integration constant. The high temperature also reduces the viscosity, while providing more kinetic energy to the water vapour for transport through the membrane, that in turn increases the transmembrane flux[51]. Additionally, the mesh size $\bar{\xi}$ of the hydrogel network increases with temperature according to[52]

$$\bar{\xi} = \sqrt[3]{\frac{6\lambda RT}{\pi N_A G}} \qquad (5)$$

where $\lambda$ is a front factor, $N_A$ is the Avogadro's constant and $G$ is the shear modulus. Considering that the PVA hydrogel mesh size is in the range of a few nanometers[53] while the nominal pore size of the PVDF support is two orders of magnitude larger (~200 nm), it is clear that the limiting layer for mass transfer through the composite membrane is the hydrogel assuming a Knudsen-type transport mechanism[54]. Therefore, increasing the mesh size with temperature would have a positive effect on the mass transfer and transmembrane flux.

## Osmotic distillation (OD) tests

The samples were tested in thermostatic conditions by using a lab-scale osmotic distillation (OD) plant (Supplementary Fig. 2)[76]. It comprised a flat membrane module with active membrane area of 3.75 cm², a peristaltic pump, an electronic balance, and graduated cylinders for monitoring stripping and feed volume variations, respectively. The apparatus was placed in a thermostatic incubator chamber working in the temperature range 28–48 °C that includes the reported[44] phase transition temperature of the TBB crystals (39–46 °C).

In the OD system, the membrane was put in contact with a 0.5 M NaCl aqueous solution (feed) and a 35 wt.% $MgCl_2$ drying (osmotic) solution on the opposite surface. Water vapour selectively migrates through the membrane pores driven by the partial pressure gradient that has been established between the two solutions at the membrane interfaces. The TBB crystals in the hydrophilic PVA layer were put in contact with the feed solution (Supplementary Fig. 2). The hydrophobicity of the PVDF membrane prevents the transport of the liquid phase (pore wetting), increasing feed concentration over time. The membrane performance at each tested temperature was estimated via the transmembrane flux, calculated as

$$J = \frac{V}{A_m \cdot \triangle t} \qquad (6)$$

where $V$ is the volume of liquid that has passed through the membrane in the fixed time interval $\Delta t$, and $A_m$ is the effective membrane area.

The rejection to NaCl was determined by an electrical conductivity-meter (Jenway, Bibby Scientific, UK) placed on the feed reservoir. Solute rejection $R\%$ is defined as

$$R\% = \left(1 - \frac{C_{distillate}}{C_{feed}}\right) \cdot 100 \qquad (7)$$

where $C_{feed}$ and $C_{distillate}$ are the NaCl concentrations in the feed and in the distillate, respectively. $R\%$ is estimated by taking into account the electrical conductivity of the feed, the effective transmembrane flux and, after opportune calibration, mass balance.

For the fouling tests, the organic foulants were dissolved in high concentration (100 ppm BSA, 50 ppm SA and 50 ppm HA) in 0.5 M NaCl solution, and used as feed, with a 35 wt% $MgCl_2$ drying (osmotic) solution on the opposite surface. The tests were performed by incorporating the OD system comprising membranes at 1.00 mg cm⁻² loading of TBB crystals in a thermostatic box operated sequentially at different temperatures during the same experiment (28 °C for 140 min, 48 °C for 140 min, and 28 °C for 140 min), and measuring the transmembrane flux with time.

The antifouling aptitude of a P-P-T membrane with 1.0 mg cm⁻² loading of TBB (P-P-T 1.0) and unloaded membranes was assessed by multiple cycles performed sequentially on the same sample. After each test that lasted 5 hours, the feed and drying solutions were replaced with freshly prepared saline solution containing foulants (100 ppm BSA, 50 ppm SA and 50 ppm HA in 0.5 M NaCl) and 35% $MgCl_2$, respectively, while the membrane was washed by flushing with MilliQ water for 20 min at room temperature (~22 °C). Before starting each cycle, the whole system was kept at the working temperature of 48 °C in an environmental chamber for 60 min. The transmembrane flux was calculated based on Eq. 6. This was done as both total value, which accounts for the total amount of water that had passed through the active membrane area within the testing time of 5 h, and an instantaneous value, which is calculated between two consecutive sampling points taken periodically during each cycle within the 5 h of operation.

## Membrane distillation (MD) tests with hypersaline solution

The membrane comprising 1.00 mg cm⁻² loading of TBB crystals (sample P-P-T 1.0) was tested in a lab-scale DCMD plant (Supplementary Figure 8). This included a flat membrane module with active membrane area of 24 cm², a two-channel peristaltic pump, and graduated cylinders for monitoring distillate and feed volume variations, respectively. For the three consecutive tests, the same membrane was put in contact on the feed side with a freshly prepared hypersaline solutions at 228 g L⁻¹ TDS (Supplementary Table 5) containing 100 ppm BSA, 50 ppm SA and 50 ppm HA. Distilled water was used as condensing fluid on the opposite membrane side. Both solutions were circulated counter-currently at $5 \times 10^{-3}$ m s⁻¹. Water vapour selectively migrates through the membrane pores driven by the partial pressure gradient that has been established between the two solutions at the membrane interfaces under a gradient of temperature of 59 °C and 20 °C at the feed and distillate side, respectively. Similar reference tests were performed with PVDF-PVA membrane (sample P-P) without

TBB crystals. The transmembrane flux (Eq. 6) was registered over 5 hours of operation for each cycle and the rejection to salts was determined by an electrical conductivity-meter placed on the distillate reservoir, and calculated as reported before (Eq. 7). Between two consecutive tests the membrane was washed by flushing for 15 minutes with MilliQ water preheated at ~40 °C.

## Frequency-dependent capacitance-frequency measurements (C–f)

The frequency-dependent capacitance measurements (C–f) were performed using four-terminal pair configurations, which were part of the multifrequency capacitance measurement unit of the Agilent B 1505 A curve tracer and LCR Meter Keysight E4980A. The four-terminal pair setup is illustrated in Supplementary Figure. 10. The parallel-plate capacitor was constructed by sandwiching the polymer films (both pristine and with TBB crystals) between two symmetric aluminum plates, as depicted in Supplementary Figure. 11. The area of the parallel-plate capacitor was 0.5 cm × 0.5 cm (0.25 cm$^2$), and the thickness of the introduced polymer was 200 μm. All measurements were performed at a bias voltage of 20 V and a frequency sweep from 1 kHz to 1 MHz. Temperature-dependent measurements were recorded from 25 °C to 50 °C by placing the capacitor on a heating stage equipped with a heater temperature controller. Interval of at least 15 min was allowed between consecutive measurements to reach a heat transfer equilibrium-like state.

## Variable-temperature X-ray diffraction

The thermal expansion of TBB was studied by using Bruker APEX DUO diffractometer with CuKα radiation (1.5418 Å) and Photon II detector. A Cryostream (Cryostream Oxford Cryosystems, Oxford, United Kingdom) was used for data collection from 275 to 310 K at a heating rate 10 K min$^{-1}$. The data was collected at 5 K intervals using the software APEX 3[77]. The data was scaled and absorption correction was applied using SADABS. The structure determination and refinement, using the OLEX2 interface[78], were performed by using the full-matrix least-squares method based on $F^2$ against all reflections with SHELXL-2014/7[79].

## Data availability

The data supporting the key findings of this study are available within the article and the Supplementary Information. Source data are provided with this paper.

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

## Acknowledgements

We thank Dr. Abdullah Khalil and Rushna Quddus, who contributed to the preliminary stage of this project. We acknowledge the BIOPUR project for partially funding this work at CNR, and New York University Abu Dhabi for the financial support. This research was partially carried out by using the Core Technology Platform resources at New York University Abu Dhabi. P.N. thanks New York University Abu Dhabi for financial support as a Research Enhancement Fund (REF). This material is based upon works supported by Tamkeen under NYUAD RRC Grant No. CG011.

## Author contributions

Conceptualization: G.D.P. and P.N.; funding acquisition: G.D.P. and P.N.; data curation: E.P., E.A., A.B., E.F., I.T., D.P.K., N.A.A., G.D.; investigation; E.P, A.B., E.A., E.F., I.T., D.P.K., N.A.A., G.D.; methodology: G.D.P. and E.P.; supervision: G.D.P., M.R. and P.N.; validation; E.P., E.A., A.B., I.T. D.P.K., N.A.A., G.D.; visualization: G.D.P., E.P., E.A.; writing (original draft): G.D.P., E.P., P.N., A.B., E.F.; writing (review & editing): G.D.P., P.N., E.P., E.A, A.B., E.F,; equal contribution: E.P and E.A.

## Competing interests

The authors declare no competing interests.
