## [Peer Review File · Nature Communications]

nature portfolio

Peer Review FileReviewer comments, first round

Reviewer #1 (Remarks to the Author):

The idea of dynamic hybrid membrane presented in this manuscript seems to be an interesting and novel approach as one of the active fouling control measures. I have two additional comments which might increase the clarity and impact of this study.

1. Water vapor transport mechanism in the temperature range 28-40 °C. The authors explained the results presented in Figure 2B that "a different transport mechanism across the membrane is activated, likely due to the change in physical state of the dynamic crystals in response to heating". However, this different transport mechanism is read ambiguous. More specifically, what kinds or types of different transport mechanisms can take place under the employed relatively moderate temperature range of 28-40 °C?

Also, the authors assert that "the deviation of increasing trend in water vapor flux was observed near the phase transition region". However, it is likely that the observed deviation has something to do with the exponential increases of water vapor pressure as a function of temperature. Did the authors happen to calculate the vapor pressure difference between the 0.5 M NaCl and 35%wt MgCl₂ solutions across the temperature range evaluate?

2. While I understand that this is a proof-of-concept study, it would be beneficial for readers to elaborate the true potential of this concept and technology. This smart dynamic hybrid membrane is designed to be employed in osmotic distillation process, which showed more than an order of magnitude lower water permeability and additional heat input ~50 °C (i.e., temperature employed for cycling performance test in Figure 4) seems considerable energy consumption compared to those of the state-of-the art reverse osmosis (RO) desalination technology. Can authors elaborate in which aspects this dynamic hybrid membranes outperform RO technology?

Reviewer #2 (Remarks to the Author):

This work describes a novel approach for developing a responsive and smart gating membrane where stimuli-responsive organic crystals were incorporated into the surface layer of a static membranes that respond to heat. The preparation of hybrid membranes involved casting a thin layer of hydrogel made of polyvinyl alcohol that contains randomly dispersed and oriented thermosensitive crystals (1,2,4,5-tetrabromobenzene) over the surface of the porous polyvinylidene fluoride (PVDF) membranes. The resulting smart membranes were evaluated for their ability to remove foulants and increase mass transfer during osmotic distillation experiments. The crystal's rapid mechanical response such as sudden expansion or motion on the surface of the membrane under the thermal stimulation slightly above the room temperature activated the membrane and removed foulants effectively. The gating effect observed in the smart dynamic membranes provides a self-cleaning capability that enhanced the membrane's mass transport performance, removes foulants effectively, and extends membrane's operational lifetime.

Strengths:

1. Overall, the paper is well written and the experimental data presented in this manuscript are very clear.
2. The conclusion is well supported by the experimental data.
3. The surface of a widely used and relatively inexpensive membrane was engineered to create a stimuli-responsive and self-cleaning membrane.
4. The development of self-cleaning and responsive membranes has potential for applications in antifouling and could potentially a significant role in addressing the challenges of water scarcity and sustainability.

Questions and comments:

The author has described and we also agree with that this approach has the potential for

developing a variety of energy-efficient self-cleaning hybrid membranes for water desalination and other separation processes. While the evaluation of the antifouling and water transport behavior of the smart hybrid membrane via osmotic distillation is an important step towards understanding its potential, it is essential to test the membrane's performance in any of the other widely-used water purification techniques and the author has also stated at line 32 that reverse osmosis accounting for over 60% of the desalination capacity worldwide. If this approach could demonstrate similar performance in other commonly-used water purification setups, it would have a much broader potential and would be of great interest to the broad research community. The challenge in that case will be the heating required for massive quantities of water, unless localized heating is used. These are the minor concerns to improve the manuscript before considering for publication.

1. The citation for magnetic or electric fields responsive membranes are missing/misplaced at the line number 52 in the introduction part.
2. Citation to few figures are missing in the manuscript. For example, Figure 1D, 1E and 1H.
3. Although the gas separation experimental data predicts about the presence of the porous structure on the surface of the hybrid membranes; however, it is not clear that why this membrane's performance is not significantly enhanced at higher temperature (above the phase transition of TS crystals) for gas separation applications as compare to lower temperature for a particular membrane. As the popping out of some TBB crystals during the phase transition could enhanced the surface porosity, which in turn could enhance the gas transport rate as similar to the water transport at higher temperature. Or is there a possibility that the TS behavior of crystals was suppressed by the transmembrane pressure that applied during the gas transport experiment?

Reviewer #3 (Remarks to the Author):

This manuscript presents a study that fabricated a PVDF membrane incorporated with stimuli-responsive organic crystals. The fabricated membranes demonstrated self-cleaning capability. I don't support this manuscript to be published on Nature Communications, because of the following reason:

In the Introduction section, the authors highlight the importance of reverse osmosis (RO). However, they selected to use a hydrophobic PVDF membrane with a pore size of 200 nm, which are several orders of magnitude larger than the "pore" size of RO membranes. Therefore, the value of this study is not justified. The reviewer is aware that osmotic distillation is used for desalination in this study. However, the water flux of this technology shown in this study ($\sim 1 \text{ L m}^{-2} \text{ h}^{-1}$) is much lower than those typically achieved by RO ($> 50 \text{ L m}^{-2} \text{ h}^{-1}$) or other membrane technologies (e.g., membrane distillation, $> 29 \text{ L m}^{-2} \text{ h}^{-1}$) at the lab scale. Therefore, I am not convinced that this study has the potential to enhance the efficiency of current desalination technologies.

Response to the comments from [redacted] Reviewers

We thank [redacted] Reviewers for their valuable comments, which have contributed significantly to improving the quality of our manuscript. We considered all comments, and in the revised version of the main text and the supplementary materials, we tried to address them to the best of our ability. Unless stated otherwise, all numbers of the figures and supplementary materials refer to the revised version of the manuscript. With this submission, we have provided marked copies of the main text and the supplementary materials where the changes to the original version have been marked. For convenience, in what follows, the original comments from the editor and reviewers are highlighted in blue color, our response is provided in black color, and the text that was added to the manuscript is marked with red color.

Response to the comments from Reviewers

Response to the comments from Reviewer #1:

Comment: *The idea of dynamic hybrid membrane presented in this manuscript seems to be an interesting and novel approach as one of the active fouling control measures. I have two additional comments which might increase the clarity and impact of this study.*

Water vapor transport mechanism in the temperature range 28-40 °C. The authors explained the results presented in Figure 2B that "a different transport mechanism across the membrane is activated, likely due to the change in physical state of the dynamic crystals in response to heating". However, this different transport mechanism is read ambiguous. More specifically, what kinds or types of different transport mechanisms can take place under the employed relatively moderate temperature range of 28-40 °C?

Response to the comment: This is an important and astute point raised by the Reviewer, and we are glad it was brought to our attention. In order to clarify the water transport mechanism, we have performed several types of experiments that were aimed to investigate the changes in the membrane that occur across the phase transition temperature. The techniques we used to that end include scanning electron microscopy (SEM), measurement of the dielectric properties, the zeta potential, the isoelectric point, and wettability via contact angle measurements. These results from these experiments were indeed very useful to determine the porosity, polarity, surface charges and hydrophilicity of the pristine and doped membranes. Specifically, it was concluded that membranes loaded with TBB crystals show enhanced hydrophilicity, improved porosity and higher isoelectric point, while they also have lower zeta potential and reduced contact angle, characteristics that are the ideal for enhanced water permeability and salt rejection (Figures 2 and 4, and references 1–9)

Moreover, additional experiments were carried out to study the dielectric properties of pristine and doped membranes using electrochemical techniques which are otherwise widely used to obtain information on the polarized membrane behavior (references 10–12), and it was observed that the doped membranes have higher dielectric constant ($\epsilon' = 2.74 \pm 0.04$) compared to the pristine membrane ($\epsilon = 2.06 \pm 0.06$) (see new Supplementary Figure 9). The dielectric constant of the doped membrane was found to be 25 % higher than that of the pristine membranes. Consequently, the incorporation of TBB crystals in the polymer matrix has resulted in increased porosity, polarity, hydrophilicity, surface charge, and the value of the dielectric parameter. These parameters are consistent and supportive to explain the increased water flux of the doped membrane from a mechanism perspective.

Moreover, gas permeability was also studied for H₂, N₂ and CO₂ as a function of the temperature, and the results are reported in Figures 3B–F. The selectivities for H₂/N₂ and H₂/CO₂ gas combinations were found to be close to the Knudsen selectivity values (see Supplementary Table 2; reference 13). This result confirms the porous structure in the selective layer of all hybrid membranes, with most of the pores having diameters between 2 and 50 nm. For this reason, while gas selectivity effects were masked by the presence of large pores which dominate the transport through the membrane, the deviation from an Arrhenius-type mass transport curve across the transition region (Figure 2D) is clearly due to the dynamic effect of the TBB crystals in response to heating.

In order to quantify any thermal expansion which could be responsible for the switching of some of the physico-chemical properties of the TBB crystals, especially its thermal expansion, we also performed variable temperature single crystal X-ray diffraction analysis on a TBB crystal. A slight shift in the diffraction pattern was observed during the diffraction experiments from 275 to 310 K (before the phase transition occurs), suggesting measurable thermal expansion in the crystal lattice. Furthermore, by using the diffraction data collected from 275 to 310 K, the thermal expansion coefficients were calculated by employing the PASCAL program for calculations and visualization of the thermal indicatrices developed by M. Lertkiattrakul and M. Cliffe based on its original version (reference 14). It was found that between 275 K and 310 K, just before the phase transition, TBB crystals show a negative thermal expansion along the principal X₁ axis [0.661, 0.0, 0.7504] with values of the thermal expansion coefficient of -358.03 MK^{-1} , and also along X₂ axis [0.9976, -0.0, -0.069] with a value of -194.62 MK^{-1} , while positive thermal expansion occurs along the X₃ axis [-0.0, 1.0, 0.0] with a coefficient of 275.15 MK^{-1} (new Supplementary Figure 12, and new Supplementary Tables 6–9). Overall, we conclude that the thermal expansion in the crystal lattice of TBB is highly anisotropic preceding the phase transition, and this anisotropic negative and positive thermal expansion behavior along different directions in the crystal lattice is likely to be responsible for the increased transport properties of the TBB-doped membrane.

In order to clarify this point, the following text has been added in the main text:

“In order to assess the effect of thermal expansion of TBB at temperatures below the phase transition, we analyzed the structure of TBB by using variable-temperature single crystal X-ray diffraction. A slight shift in the diffraction peak position was observed between 275 to 310 K before the phase transition, consistent with thermal expansion of the crystal lattice. The thermal expansion coefficients were determined by diffraction between 275 to 310 K.^[72] Before the phase transition, the TBB crystals undergo negative thermal expansion along the principal X₁ axis [0.661, 0.0, 0.7504] with larger coefficient of -358.03 MK^{-1} and along the X₂ axis [0.9976, -0.0, -0.069] with -194.62 MK^{-1} . This is compensated with a positive thermal expansion along the X₃ axis [-0.0, 1.0, 0.0] with a coefficient of 275.15 MK^{-1} (Supplementary Figure 12, Supplementary Tables 6–9). The strongly anisotropic thermal expansion is expected to affect the contact between the crystals and the matrix, and could lead to enhanced transport properties of the membrane. Additional experiments were carried out to study the dielectric properties of pristine and doped membranes using electrochemical techniques which are widely used to characterize the polarized membrane behavior.^[73–75] It was observed that the doped membrane has 25% higher dielectric constant ($\epsilon' = 2.74 \pm 0.04$) compared to the pristine membrane ($\epsilon = 2.06 \pm 0.06$) (Supplementary Figure 9).”

The new figures are provided below for Reviewer’s perusal:

New Supplementary Figure 9. Dielectric properties as a function of the frequency at 25 °C and at 50 °C for pristine PVDF-PVA (P-P) membrane and doped PVDF-PVA-TBB (P-P-T) membrane.

Supplementary Figure S12. Plots of the expansivity indicatrices along the principal axes (units: MK^{-1}) for TBB crystal.

Comment: *Also, the authors assert that "the deviation of increasing trend in water vapor flux was observed near the phase transition region". However, it is likely that the observed deviation has something to do with the exponential increases of water vapor pressure as a function of temperature.*

Response to the comment: We have carefully considered the suggestion by the Reviewer, and tried to find evidence for such hypothesis. However, our experimental results, unfortunately, do not support this line of thought. The exponential dependence of the vapor pressure on the temperature (e.g. through

the Antoine equation)—directly related to the flux—is already included in the linearization of the flux equation that is shown in Figure 2D. Therefore, the deviation of the experimental data from linearity implies that factors other than those accounted for by the (simple) Antoine equation would be needed to describe the transport of vapor beyond the transition temperature that should be related to the “improved” or, at least, “different” transport mechanism. This has been addressed in greater detail in our response to the previous comment.

Comment: *Did the authors happen to calculate the vapor pressure difference between the 0.5 M NaCl and 35%wt MgCl₂ solutions across the temperature range evaluate?*

Response to the comment: Data on the vapor pressure for 0.5 M NaCl and 35% wt MgCl₂ solutions across the relevant temperature range is reported in the figure below, that was taken from the following reference: R. H. Perry, D. W. Green, Perry's Chemical Engineers' Handbook, 8th ed., McGraw-Hill, 2007 (ISBN 0-07-142294-3). Since the difference in vapor pressure between the two solutions increases by only 0.1 kPa on going from 44 to 48 °C (from 8.3 to 9 kPa at 44°C and from 10.2 to 11 kPa at 48 °C for 0.5 M NaCl and 35% wt. MgCl₂, respectively), the data convincingly confirms that abrupt changes in vapor pressure difference between the feed and the stripping solutions in OD configuration cannot be taken as the reason for the deviation from the Arrhenius-like curve in Figure 2D.

Comment: *While I understand that this is a proof-of-concept study, it would be beneficial for readers to elaborate the true potential of this concept and technology. This smart dynamic hybrid membrane is designed to be employed in osmotic distillation process, which showed more than an order of magnitude lower water permeability and additional heat input ~50 °C (i.e., temperature employed for cycling performance test in Figure 4) seems considerable energy consumption compared to those of the state-of-the art reverse osmosis (RO) desalination technology. Can authors elaborate in which aspects this dynamic hybrid membranes outperform RO technology?*

Response to the comment: As is has been well noted by the Reviewer, our work presented in this manuscript aims to demonstrate that: 1) the flux can be improved, and, 2) the membranes based on hybrid material described here are less prone to fouling, with the functional layer containing the thermally active (thermosalient) TBB crystals being the key to this technology.

In order to demonstrate the effect of the transition temperature, we investigated the membranes in the process of osmotic distillation (OD), which is a thermostatic process by operating at different temperatures across the transition region. In a further extension of this work, the same membranes could also be used in membrane distillation (MD), which is a thermally activated process and whose working mechanism is similar to that of the OD in the sense that a gradient of vapor pressure across the membrane is the driving force of both processes—for OD this gradient is generated by a difference in solute concentration between the feed and the stripping solutions while in case of the MD process, it is generated by the difference in temperature between the two sides. Therefore, having proved the applicability of the concept that we propose, in principle, TBB-loaded membranes or similar hybrid materials can be also used in MD. This is a technology that has been frequently investigated for treatment of hypersaline solutions whose osmotic pressure extends beyond the capabilities of the reverse osmosis (RO). Indeed, while RO is suitable to recover pure water from salty solutions up to 70-90 g/L TDS, due to the high hydraulic pressure needed to overcome the osmotic pressure that might have irreversible impacts on membrane permeability and selectivity, the feed concentration has a relatively small effect on the mass flux for MD. Therefore, despite the traditionally lower transmembrane flux of MD compared to RO, the two technologies practically differ in their range of applicability with respect to the feed solution composition. RO is energetically viable to desalt water but it cannot be used at all when the feed concentration exceeds certain threshold values. On the other hand, MD being energetically more expensive than RO, can be operationally applied up to the saturation of the feed; membrane crystallization is just an extension of MD in which the same membranes can be used. Furthermore, since suitable transmembrane fluxes can be generated under relatively low feed temperatures (e.g. 59 °C as in the present study), MD can be operated in combination with other industrial processes with waste heat or even by using solar energy. Therefore, the interest in these membranes would be for the range where RO is not applicable, i.e. for hypersaline solutions and with low-grade heat.

Response to the comments from Reviewer #2:

Comment: *This work describes a novel approach for developing a responsive and smart gating membrane where stimuli-responsive organic crystals were incorporated into the surface layer of a static membranes that respond to heat. The preparation of hybrid membranes involved casting a thin layer of hydrogel made of polyvinyl alcohol that contains randomly dispersed and oriented thermosolient crystals (1,2,4,5-tetrabromobenzene) over the surface of the porous polyvinylidene fluoride (PVDF) membranes. The resulting smart membranes were evaluated for their ability to remove foulants and increase mass transfer during osmotic distillation experiments. The crystal's rapid mechanical response such as sudden expansion or motion on the surface of the membrane under the thermal stimulation slightly above the room temperature activated the membrane and removed foulants effectively. The gating effect observed in the smart dynamic membranes provides a self-cleaning capability that enhanced the membrane's mass transport performance, removes foulants effectively, and extends membrane's operational lifetime.*

Strengths:

- 1. Overall, the paper is well written and the experimental data presented in this manuscript are very clear.*
- 2. The conclusion is well supported by the experimental data.*
- 3. The surface of a widely used and relatively inexpensive membrane was engineered to create a stimuli-responsive and self-cleaning membrane.*

4. The development of self-cleaning and responsive membranes has potential for applications in antifouling and could potentially play a significant role in addressing the challenges of water scarcity and sustainability.

Response to the comment: We thank the Reviewer for the very thorough assessment of the results presented in our manuscript, and we thank them for highlighting the main strengths of the material presented here.

Comment: *Questions and comments: The author has described and we also agree with that this approach has the potential for developing a variety of energy-efficient self-cleaning hybrid membranes for water desalination and other separation processes. While the evaluation of the antifouling and water transport behavior of the smart hybrid membrane via osmotic distillation is an important step towards understanding its potential, it is essential to test the membrane's performance in any of the other widely-used water purification techniques and the author has also stated at line 32 that reverse osmosis accounting for over 60% of the desalination capacity worldwide. If this approach could demonstrate similar performance in other commonly-used water purification setups, it would have a much broader potential and would be of great interest to the broad research community.*

Response to the comment: We thank again the Reviewer for the very positive assessment of the results presented in our article. We also appreciate that the Reviewer recognizes the relevance of the results presented in this manuscript. Our intention in a future development of this technology through processes of optimization and selection would be, indeed, to expand the application to other processes that are used for desalination, especially because the concept is general in that describes a combination of two very different classes of engineering materials (polymers and crystals). Based on our extensive experience, particularly of one of the contributing research groups, in materials chemistry for desalination, however, we feel that a single type membrane would be extremely challenging to be applied indifferently in other commonly-used water purification setups, while anticipating that it will show a similar or comparable performance due to the very different nature of the underlying physical principles. Specifically, we would like to highlight that the membranes that we describe here are developed for OD, MD or MCr (membrane crystallization). On the other hand, the reverse osmosis (RO) capitalizes on dense (no pores), asymmetric, thin film composite membranes that are normally made from hydrophilic polymers. Importantly, the working mechanism is quite different: high hydrostatic pressure (>50 bar) for a transport based on solution diffusion (of water in the polymer) mechanism, and this comes with the limitation outlined above. Placing this into the perspective of future modifications to explore the idea proposed by the Reviewer, we could envisage an approach whereby the surface of an RO membrane is functionalized with the PVA-TBB layer, and the resulting membranes could be assessed for the performance. Given that such an endeavor would imply the development of a completely and substantially different class of membranes and would make a full-fledged research project, we would like to leave this into the perspective of a follow-up work to the results described in the present manuscript.

Comment: *The challenge in that case will be the heating required for massive quantities of water, unless localized heating is used.*

Response to the comment: We appreciate the amount of detail to which the Reviewer has attended to, and we are pleased to see that this comment is somewhat consistent with tone of the comments from Reviewer #1. In response to this comment, we would like to reiterate that with this research project we were hoping to accomplish two major improvements in the performance of the membranes that are currently available. These two major pinnacle goals were improved flux, and decrease of the proneness of the membranes to fouling.

Being encouraged by the comments from both reviewers, we undertook a series of experiments aimed to clarify the effect of the phase transition temperature. These experiments were performed by resorting to the process of osmotic distillation (OD), which is by definition a thermostatic process, while operating at different temperatures. We note that within a perspective of future expansion of this work, the same membranes could be used in membrane distillation (MD), a thermally activated process whose working mechanism is similar to that of the OD (in the sense that a gradient of vapor pressure across the membrane is the driving force of both process). Having demonstrated the concept we propose, we suggest that TBB-loaded membranes could be used in MD. We note that this technology has been well established for the treatment of hypersaline solutions whose osmotic pressure lies beyond those typical for reverse osmosis (RO). Indeed, we note that while RO is suitable to recover pure water from salty solutions up to 70–90 g/L TDS, due to the high hydraulic pressure needed to overcome the osmotic pressure that might have irreversible impacts on membrane permeability and selectivity, the feed concentration has a minor effect on the mass flux in case of MD. Therefore, despite the traditional lower transmembrane flux of MD relative to RO, the two technologies are different in the range of applicability with respect to the feed solution composition. RO is energetically viable to desalt water but remains inapplicable when the feed concentration exceeds certain values. On the other hand, MD, being energetically more expensive than RO, can be operationally applied up to the saturation of the feed (indeed, the membrane crystallization is an extension of MD in which the same membranes can be used). Furthermore, since suitable transmembrane fluxes can be generated under relatively low feed temperatures (e.g. 59 °C as in the present study), MD can be operated in combination with other industrial processes with waste heat or by using solar energy. Therefore, the interest in these membranes would be for the range where RO is not applicable, i.e. for hypersaline solutions and with low-grade heat.

Comment: *These are the minor concerns to improve the manuscript before considering for publication. 1. The citation for magnetic or electric fields responsive membranes are missing/misplaced at the line number 52 in the introduction part.*

Response to the comment: The suggested citations have now been corrected at page 3.

Comment: *2. Citation to few figures are missing in the manuscript. For example, Figure 1D, 1E and 1H.*

Response to the comment: The suggested figures have now been cited in the main text.

Comment: *3. Although the gas separation experimental data predicts about the presence of the porous structure on the surface of the hybrid membranes; however, it is not clear that why this membrane's performance is not significantly enhanced at higher temperature (above the phase transition of TS crystals) for gas separation applications as compare to lower temperature for a particular membrane. As the popping out of some TBB crystals during the phase transition could enhanced the surface porosity, which in turn could enhance the gas transport rate as similar to the water transport at higher temperature. Or is there a possibility that the TS behavior of crystals was suppressed by the transmembrane pressure that applied during the gas transport experiment?*

Response to the comment: In response to the Reviewer's comment, the text on page 13 has been revised to further elaborate the discussion as follows:

Lines 312-315: *"In porous structures, an increase in temperature typically reduces the permeation of gases. Specifically, when the transport is dominated by the Knudsen mechanism, the gas flux is inversely proportional to the square root of the temperature (Eqs. 3 and 4 in the Supplementary Methods). When the transport is controlled by a viscous flow, the negative effect of temperature is due to an increase in gas viscosity."*

Lines 325-329: “The ideal selectivities measured on the tested P-P-T membranes did not vary significantly with the TBB particles loading, ranging from 2.8 to 3.9 and from 2.2 to 3.8 for H₂/N₂ and H₂/CO₂ gases, respectively, confirming that gas transport through these membranes is mainly governed by the Knudsen mechanism, with some contribution from viscous flow, which explains the lower selectivities compared to the ideal Knudsen values.”

Lines 333-336: “..., the observed increase in gas permeability with temperature in P-T-T membranes (compared to reference membranes) can be attributed to the dynamic effect of TBB crystals responding to heating, which counteracts the typical negative temperature dependence of Knudsen and viscous flow mechanisms.”

Response to the comments from Reviewer #3:

Comment: *This manuscript presents a study that fabricated a PVDF membrane incorporated with stimuli-responsive organic crystals. The fabricated membranes demonstrated self-cleaning capability. I don't support this manuscript to be published on Nature Communications, because of the following reason:*

In the Introduction section, the authors highlight the importance of reverse osmosis (RO). However, they selected to use a hydrophobic PVDF membrane with a pore size of 200 nm, which are several orders of magnitude larger than the "pore" size of RO membranes. Therefore, the value of this study is not justified.

Response to the comment: We thank and appreciate the Reviewer for their assessment and the critical comments. We have discussed extensively the Reviewer's comments, and we performed additional experiments in order to address their concerns and suggestions. We are pleased to report that our experiments returned encouraging results and we hope to be able to respond to all comments to the reviewer's satisfaction. Specifically, with regards to the Reviewer's general comment, we would like to clarify that in our experiments we used the new membrane for desalination by MD or OD. Our results clearly showed that the rejection to the salt was >99%, despite the pores were about 200 nm, and this is evidenced by the experimental results. Having said that, we do concur with the Reviewer that such size of the pores might not be suitable for using the membrane described here to “filtrate” seawater under pressure (e.g. in microfiltration), where the selectivity would be practically zero.

Comment: *The reviewer is aware that osmotic distillation is used for desalination in this study. However, the water flux of this technology shown in this study (~1 L m⁻² h⁻¹) is much lower than those typically achieved by RO (>50 L m⁻² h⁻¹) or other membrane technologies (e.g., membrane distillation, >29 L m⁻² h⁻¹) at the lab scale. Therefore, I am not convinced that this study has the potential to enhance the efficiency of current desalination technologies.*

Response to the comments: We do appreciate the Reviewer's concerns, and we have considered their comment and tried to address it to the best of our ability. We would like to highlight here that in order to address this comment (as well as those from Reviewers #1 and #2) we performed additional experiments that utilized TBB-loaded membranes in MD configuration at a temperature of 60–80 °C (i.e. beyond the transition region) and by using a hypersaline solution (220 g/L TDS) as feed. The feed solution contained intentionally added foulant molecules. The results were compared with those obtained by using membranes without TBB (that is, only PVDF + PVA). Below this temperature, the driving force would not be sufficient to drive the process with this high-concentrated feed solution.

In order to respond to this comment, the following text and a figure describing the new experiments was added to the main text of the manuscript:

“One of the membrane compositions (P-P-T 1.0) was tested in consecutive cycles of direct contact membrane distillation (DCMD) experiments using hypersaline feed solution containing solute salts at 228 g/L TDS and foulant molecules (Supplementary Table 5, Supplementary Figures 7 and 8). The feed was warmed up to ~ 59 °C and thus beyond the transition temperature of TBB, and the condensing water at the distillate side was set at ~ 20 °C. Figure 5A depicts the average transmembrane flux measured for both TBB-loaded and nascent membranes during the consecutive DCMD cycles using the same membrane, with intermediate flushing of 15 minutes with distilled water pre-heated to 40 °C. Despite the reduced driving force of the process ($\Delta T = 39$ °C) and the poor fluid-dynamic conditions (Reynold number $Re = 24$, assuming a solution density of 1151.5 kg m^{-3} at 59 °C) that have minor effect on heat and mass transfer towards the membrane boundary layer,^[70] in all cycles J was found to be over $1 \text{ L h}^{-1} \text{ m}^{-2}$ for TBB-doped membrane, with a decrease of about 7% from the first to the third cycle. At such high salinity of the feed, which is well beyond the range of operability of reverse osmosis for water desalination due to osmotic pressure limitations,^[71] the thermal energy consumption with P-P-T 1.0 membrane was around 9500 kWh m^{-3} and it remained nearly constant for several cycles (Figure 5B). As comparison, for undoped PVDF-PVA membrane, the decrease in J was as high as 40% while the thermal energy consumption increased by almost 60% from the first to the third cycle.”

Figure 5. (A) Trans-membrane flux for unloaded PVDF-PVA (P-P) and loaded PVDF-PVA-TBB with 1.0 mg cm^{-2} (P-P-T 1.0) membranes in multi-cycle DCMD tests with hypersaline feed solutions and foulants. The membranes were flushed for 15 minutes with pre-heated water at ~ 40 °C between two consecutive runs. (B) Specific thermal energy consumption (STEC) for the DCMD tests.

References used in the response to the reviewers' comments:

1. Vatanpour, V. & Zoqi, N. Surface modification of commercial seawater reverse osmosis membranes by grafting of hydrophilic monomer blended with carboxylated multiwalled carbon nanotubes. *Appl. Surf. Sci.* **396**, 1478–1489 (2017).
2. Kang, G. S., Baek, Y. & Yoo, J. B. Relationship between surface hydrophobicity and flux for membrane separation. *RSC Adv.* **10**, 40043–40046 (2020).
3. Xu, at. al., Enhancing the permeability and anti-fouling properties of a polyamide thin-film composite reverse osmosis membrane via surface grafting of L-lysine. *RSC Adv.* **9**, 20044–20052, (2019).

4. Chen, L., Xu, P. & Wang, H. Interplay of the Factors Affecting Water Flux and Salt Rejection in Membrane Distillation: A State-of-the-Art Critical Review. *Water* **12**, 2841 (2020).
5. Mozia, S., Grylewicz, A., Zgrzebnicki, M., Darowna, D. & Czyżewski, A. Investigations on the Properties and Performance of Mixed-Matrix Polyethersulfone Membranes Modified with Halloysite Nanotubes. *Polymers* **11**, 671 (2019).
6. Kimani, et. al., The influence of feedwater pH on membrane charge ionization and ion rejection by reverse osmosis: An experimental and theoretical study. *J. Membr. Sci.* **660**, 120800 (2022).
7. Qin, J-J., Oo, M. H. & Coniglio, B. Relationship between feed pH and permeate pH in reverse osmosis with town water as feed. *Desalination* **177**, 267–272 (2005).
8. Mehrabi, et. al., Enhanced negative charge of polyamide thin-film nanocomposite reverse osmosis membrane modified with MIL-101(Cr)-Pyz-SO₃H. *J. Membr. Sci.* **664**, 121066 (2022).
9. Halleb, A., Nakajima, M., Yokoyama, F. & Neves, M. A. Effect of Surfactants on Reverse Osmosis Membrane Performance. *Separations* **10**, 168 (2023).
10. Li, Q., Zhao, K., Liu, Q. & Wang, J. Desalination behavior analysis of interior-modified carbon nanotubes doped membrane by dielectric spectrum and molecular simulation. *Nanotechnology* **31**, 315705 (2020).
11. Antony, A., Chilcott, T., Coster, H. & Leslie, G. In situ structural and functional characterization of reverse osmosis membranes using electrical impedance spectroscopy. *J. Membr. Sci.* **425–426**, 89–97 (2013).
12. Chang, K. & Geise, M. G. Dielectric Permittivity Properties of Hydrated Polymers: Measurement and Connection to Ion Transport Properties. *Ind. Eng. Chem. Res.* **59**, 5205–5217 (2020).
13. Giorno, L., Drioli, E. & Strathmann, H. *Encyclopedia of Membranes* (Springer, Heidelberg, 2016).
14. Cliffe, M. J. & Goodwin, A. L. *J. Appl. Cryst.* **45**, 1321–1329 (2012).

Reviewer comments, second round

Reviewer #1 (Remarks to the Author):

The authors carefully reviewed the comments and appropriately addresses some points that further strengthened the overall clarity of the paper.

Reviewer #2 (Remarks to the Author):

The authors have done an excellent job responding to reviewer comments and I am very satisfied with their responses. I have no further comments on the paper.

Reviewer #3 (Remarks to the Author):

I read the authors' response to my comments. I don't think the authors address my comments adequately. Especially, the authors did not answer about the low water permeability of the membranes in this study. Although fouling resistance is a key parameter that demonstrates the performance of the membrane, the water permeability is equally, or sometimes more important. Considering the low water permeability of the membranes (1 LMH vs. 20-30 LMH of commercial membranes), I am not convinced that the reported membrane is valuable to enhance the efficiency of desalination. Also, the authors mentioned that the thermal energy consumption with P- P-T 1.0 membrane was around 9500 kWh m⁻³ . The energy consumption is orders of magnitude higher than current brine treatment technologies. This adds my question about the value of the developed membrane. Additionally, the authors use 50-100 ppm of organic foulants for membrane distillation (MD) experiments. To my knowledge, such concentrations of foulants usually do not cause decrease of membrane performance, due to the low fouling potential of MD. The authors need to compare their membranes with commonly used membranes for MD (which have higher water permeability and likely not been fouled by organic foulants) to show the value of their membranes.

Response to the comments from [redacted] Reviewers

[reacted] We considered all comments, and we tried to address them to the best of our ability. Unless stated otherwise, all numbers of the figures and supplementary materials refer to the *revised* version of the manuscript. Together with this submission we have provided marked copies of the main text and the supplementary materials where the changes to the original version have been marked. For convenience, in what follows, the original comments from the editor and reviewers are highlighted in *blue color*, our response is provided in *black color*, and the text that was added to the manuscript is marked with *red color*.

Response to the comments from Reviewers

Reviewer #1 (Remarks to the Author):

The authors carefully reviewed the comments and appropriately addresses some points that further strengthened the overall clarity of the paper.

Response to the comment: We thank the Reviewer for the positive assessment of our efforts which we put to revised our article in response to his/her comments.

Reviewer #2 (Remarks to the Author):

The authors have done an excellent job responding to reviewer comments and I am very satisfied with their responses. I have no further comments on the paper.

Response to the comment: We are grateful to the Reviewer for their assessment and we thank them for their comments.

Reviewer #3 (Remarks to the Author):

Comment: *I read the authors' response to my comments. I don't think the authors address my comments adequately. Especially, the authors did not answer about the low water permeability of the membranes in this study. Although fouling resistance is a key parameter that demonstrates the performance of the membrane, the water permeability is equally, or sometimes more important. Considering the low water permeability of the membranes (1 LMH vs. 20-30 LMH of commercial membranes), I am not convinced that the reported membrane is valuable to enhance the efficiency of desalination.*

Response to the comments: We thank the Reviewer for their additional input. We would like to clarify that the well-known water flux in OD/MD (and in other membrane processes) is a function of both the membrane properties and process conditions. When referring to low or high water permeability of a certain membrane, this should be probably put into the context of the actual operating conditions.

In the case of our OD tests, unfortunately, there are no relevant literature data to be able to compare the performance of our membranes, since analogous working conditions have been not explored prior to this work, and therefore, commercially available counterparts simply do not exist for us to be able to compare the transmembrane flux. In the case of our DCMD tests, we used a hypersaline feed solution of 228 g/L TDS (seawater is 35 g/L), a fluid dynamic regime with Reynold's number around 25 (solutions circulation velocity of 0.005 m/s), and a temperature gradient between feed and distillate of 39 °C. Under such conditions, it is improbable that commercial membranes could provide fluxes as high as 20-30 LMH.

However, in order to address this comment, by considering the permeability instead of the flux, we calculated the overall membrane mass transfer coefficient for our best-performing membrane (please see the new revised Figure 5), since this parameter allows evaluation of the transport properties of the membrane itself, aside of the driving force. Our measurement found it to be about $2.6\text{--}2.7 \times 10^{-8} \text{ kg m}^{-2} \text{ s}^{-1} \text{ Pa}^{-1}$, values that are comparable to the data reported in the literature for commercially available polypropylene membranes (for example, see: M.-C. Sparenberg, B. Hanot, C. Molina-Fernández, P. Luis, Experimental mass transfer comparison between vacuum and direct contact membrane distillation for the concentration of carbonate solutions, *Separation and Purification Technology* **275**, 2021, 119193).

Despite the similar mass transfer properties of our membranes compared to the literature data, and the lower transmembrane flux—since we worked in different operating conditions to be able to harness the dynamic properties of the thermosolient crystals—we have clearly demonstrated the self-cleaning ability of the membranes in the presence of foulants, which was, indeed, the principal aim of our work, as is also stated in the title of the manuscript. We believe that this result is also important to enhance the efficiency of desalination in conditions of feed salinity that are outside the range of operability of reverse osmosis.

In order to address the Reviewer's comment, the following text and the revised Figure 5 have been added in the main text of the manuscript:

“However, since the flux depends on the applied driving force, the overall membrane mass transfer coefficient B_m was calculated^[70] to enable a more direct comparison between TBB-doped and undoped membranes. Indeed, B_m allows evaluation of the real permeability performance of the membrane, apart from the effect of the driving force.^[72] Figure 5A shows a steeper decrease in the mass transfer coefficient with membrane reuse for undoped membranes, which means that there is an increased mass transfer resistance due to accumulation of foulants or decrease in membrane permeability, that is

less effective for TBB-loaded samples. These data clearly demonstrate that the transmembrane flux is directly related to the intrinsic mass transfer properties of the membrane itself, independent of the effect of the driving force which is identical across all tests. Furthermore, as the membrane is reused over multiple cycles, the mass transfer properties of the P-P-T 1.0 membrane indicate that it is less prone to fouling, demonstrating the self-cleaning properties of the TBB-loaded membranes.”

Figure 5 has been revised, and the revised version is shown below:

Revised Figure 5. Multiple cycling DCMD tests of the membranes. (A) Transmembrane flux (J) and overall mass transfer coefficient (B_m) for unloaded PVDF-PVA (P-P) membrane and membra PVDF-PVA-TBB loaded with 1.0 mg cm^{-2} (P-P-T 1.0) in multiple cycles of DCMD tests with hypersaline feed solutions at 228 g L^{-1} TDS and foulant molecules as feed. The membranes were flushed for 15 minutes with pre-heated water at $\sim 40 \text{ }^\circ\text{C}$ between two consecutive runs. (B) Specific thermal energy consumption (STEC) for the DCMD tests.

We also revised our Conclusions section to reflect these results and the conclusions:

“In summary, we described a conceptually new approach to smart gating membranes where dynamic functionality is added to otherwise static membranes by surface coating with polymer-embedded TS crystals. We demonstrated that the smart membranes are capable of gating, whereby the mechanical instability of the composite, driven by the dynamic response of the crystals to heating slightly above room temperature, allows a pure water flux increase exceeding 40% for optimal TBB loading conditions in saltwater desalination by OD, compared to reference membranes. Furthermore, the inclusion of TBB crystals in the crosslinked PVA network provides an added value of enhanced fouling resistance, imparted by the enhanced hydrophilicity and the phase transition property of dynamic crystals in both OD and MD. Since the surface charge and hydrophilic character of the functional coating is tuneable with the pH of the feed solution, the proposed approach allows a double responsive mechanism—by pH and temperature—to increase the transmembrane flux, remove foulants, and extend the membrane operational lifetime

in water desalination by DCMD under conditions that are well beyond the salinity operability range of reverse osmosis. Within a broader context, considering the existing possibility to recover the enthalpy of water vaporization by distillate condensation in multiple stages to reduce thermal energy consumption, and the availability of more than twenty TS crystalline materials and different membrane compositions, this proposed approach is the first step towards the development of a great variety of smart, energy-conserving, hybrid separation membranes. This new family of membranes are endowed with capability to respond to various stimuli while they would also alleviate or completely overcome the fouling problem simply by short-term exposure to the respective stimulus.”

Comment: *Also, the authors mentioned that the thermal energy consumption with P-P-1.0 membrane was around 9500 kWh m⁻³. The energy consumption is orders of magnitude higher than current brine treatment technologies. This adds my question about the value of the developed membrane.*

Response to the comment: Indeed, we agree with the Reviewer that the energy consumption for RO is around 5 kWh m⁻³, although within the limitations highlighted above. A thermal energy consumption as high as 9500 kWh/m³ is comparable to that of the currently used membranes for DCMD. However, we note that in the MD process we need just to warm the feed up to, e.g., 59 °C or perhaps slightly higher in commercial applications (for example, 60-80 °C), which is completely compatible with the opportunity of using waste heat or alternative energy sources, such as solar energy. In the case of RO, energy is requested to generate pumping up to 60-70 bars, that is achievable only by the direct use of electrical energy. Furthermore, energy recovery devices and multi-effect MD systems are being developed that already reduce the energy consumption to around 100-200 kWh/m³ and it is very likely that this value will decrease in the near future. In our case, the energy consumptions have been reported to show that despite the high absolute value, with TBB-doped membranes the energy requirements are almost the same in several cycles, while for undoped membranes the energy consumption constantly increases due to the reduced productivity (less amount of desalinated water) and to progressive wasting of membrane permeability. We would like to stress here that—at least at the present stage of proposed concept and without actual optimization—our proposed membranes should not be thought of as an alternative, but rather as complementary to the conventional RO membranes, since they can be used in the conditions that are not applicable to RO.

Comment: *Additionally, the authors use 50-100 ppm of organic foulants for membrane distillation (MD) experiments. To my knowledge, such concentrations of foulants usually do not cause decrease of membrane performance, due to the low fouling potential of MD. The authors need to compare their membranes with commonly used membranes for MD (which have higher water permeability and likely not been fouled by organic foulants) to show the value of their membranes.*

Response to the comment: We thank the Reviewer for this astute observation. We would like to highlight that such organic foulants and concentrations are reported in many articles on MD to study the fouling aptitude of the DCMD membranes. The three

molecules are representative of the main class of foulants that can be found in natural feed solutions: proteins, carbohydrates and humic substances. Importantly, these substances might cause direct fouling on the membrane surface, thus reducing water permeability due to increased resistance to mass transfer. More importantly, they could create hydrophilic regions that will provide membrane wetting that, in turn, will induce the flux to collapse. Actually, we observed that in the used conditions, undoped membranes underperform when they are used in several cycles, due to fouling, which means that such foulants are effective for these membranes although wetting was not yet observed (there was no increase in conductivity on the distillate side). We hope that this helps to clarify the point of the Reviewer's concern.